# Efficient Prompt Optimization Through the Lens of Best Arm Identification

**Chengshuai Shi**[*]
University of Virginia
cs7ync@virginia.edu

**Kun Yang**[*]
University of Virginia
ky9tc@virginia.edu

**Zihan Chen**
University of Virginia
brf3rx@virginia.edu

**Jundong Li**
University of Virginia
jundong@virginia.edu

**Jing Yang**
The Pennsylvania State University
yangjing@psu.edu

**Cong Shen**
University of Virginia
cong@virginia.edu

## Abstract

The remarkable instruction-following capability of large language models (LLMs) has sparked a growing interest in automatically finding good prompts, i.e., prompt optimization. Most existing works follow the scheme of selecting from a pre-generated pool of candidate prompts. However, these designs mainly focus on the generation strategy, while limited attention has been paid to the selection method. Especially, the cost incurred during the selection (e.g., accessing LLM and evaluating the responses) is rarely explicitly considered. To overcome this limitation, this work provides a principled framework, TRIPLE, to efficiently perform prompt selection under an explicit budget constraint. TRIPLE is built on a novel connection established between prompt optimization and fixed-budget best arm identification (BAI-FB) in multi-armed bandits (MAB); thus, it is capable of leveraging the rich toolbox from BAI-FB systematically and also incorporating unique characteristics of prompt optimization. Extensive experiments on multiple well-adopted tasks using various LLMs demonstrate the remarkable performance improvement of TRIPLE over baselines while satisfying the limited budget constraints. As an extension, variants of TRIPLE are proposed to efficiently select examples for few-shot prompts, also achieving superior empirical performance.

## 1 Introduction

Large language models (LLMs) have rapidly changed technology landscapes in our society [20, 72, 73, 11, 55]. Researchers continuously find effective ways to unlock their potential on various downstream tasks. Among different research directions, the remarkable ability of LLMs to follow instructions has motivated the study of searching for suitable prompts to interact with them [50]. This approach is particularly attractive as it does not require updating the inside parameters of an LLM, and is natural in the way of human conversations. Nevertheless, it has also been recognized that the performance of an LLM is sensitive to the selected prompts [87, 52], and manually designing suitable prompts can be a labor-intensive process [54]. Thus, there is a growing interest to perform automatic prompt optimization [90, 79, 21, 19, 59, 84, 28, 58, 60, 81].

While these studies have proposed different prompt optimization designs, they commonly follow the approach of generating a pool of candidate prompts and then selecting from them. With a deeper look, it can be recognized that the focus in these existing works largely leans towards how to generate the candidate pool, while limited attention has been paid towards how to select from the candidates.

---

[*]indicates equal contributions, random order.

38th Conference on Neural Information Processing Systems (NeurIPS 2024).

For example, many works [35, 79, 28, 59] directly evaluate all the generated prompts on the entire development dataset. However, this less-emphasized selection process typically requires accesses to LLMs, which are often (1) *financially costly* (e.g., each OpenAI API access incurs a cost); (2) *time-wise consuming* (e.g., even a locally hosted LLM would typically require seconds to respond); (3) under *total usage limits* (e.g., OpenAI has hard per-day and per-month limits on API accesses). Furthermore, it is often overlooked that evaluating the responses of an LLM for different candidate prompts can be costly as many tasks (e.g., writing improvement, mathematical reasoning, etc.) would require human (and sometimes domain expert) opinions. As a result, the prompt optimization process can incur an unaffordable cost without a proper selection method.

To make the learning process more accessible, this work proposes to study prompt optimization under an explicitly imposed budget constraint when interacting with the targeted LLM, in addition to the previously considered requirements (e.g., discrete, interpretable, and black-box). To the best of our knowledge, budget constraints are only briefly mentioned in Zhou et al. [90], Pryzant et al. [60], and there are no systematic or principled investigations of how to address the limited budget constraint in prompt optimization. The main contributions of this work are summarized as follows.

• The constraint of a limited budget is explicitly introduced into prompt optimization, which has been largely ignored before. As most of the prompt optimization methods rely on selecting from a pre-generated candidate prompt pool, we focus our study on how to carefully allocate budgets to test each candidate prompt so that the optimal one can be learned efficiently and effectively.

• We propose a general solution framework, termed TRIPLE (bes**T** a**R**m **I**dentification for **P**rompt **LE**arning), by establishing a novel connection between prompt optimization and multi-armed bandits (MAB) [41]. In particular, we focus on harnessing the power of fixed-budget best arm identification (BAI-FB) [4, 37] to address prompt optimization (especially, selection) with a limited budget constraint. Two representative designs TRIPLE-SH and TRIPLE-CR, inspired by celebrated BAI-FB algorithms, are presented. To improve scalability, two enhanced methods, TRIPLE-CLST and TRIPLE-GSE, are further proposed, where prompt embeddings are leveraged by exploiting the ideas of clustering and function approximation to accelerate the learning process.

• Extensive experimental results are reported using well-adopted prompt tasks and varying LLMs to demonstrate the superiority of TRIPLE over previous baselines. In particular, on GPT3.5 and Llama2, compared with baseline methods also not using prompt embeddings, the basic TRIPLE-SH and TRIPLE-CR achieves performance improvements by (on average) 3% to 16%. When leveraging prompt embeddings, the enhanced TRIPLE-CLST and TRIPLE-GSE also outperform corresponding baselines by (on average) 10% to 56% with fewer prompts than budget and (on average) 16% to 45% with more prompts than budget. The gains are further evidenced on other LLMs, i.e., Gemma and Mistral. Moreover, the proposed methods can be directly plugged into two popular prompt optimization pipelines, APE [90] and APO [60], with end-to-end performances significantly improved over their original implementations.

• This work extends broadly to providing a new perspective of prompt optimization from MAB, and also a new application scenario of MAB in prompt optimization. This established connection may spark further innovations in both fields. As one concrete example, we extend the study to optimizing the selection of examples in few-shot prompts [9], which can be recognized as a BAI-FB problem in the setup of combinatorial bandits [14, 12]. Experimental results illustrate that the extensions of TRIPLE achieve superior performance, demonstrating its rich potential.

**Key Related Works.** We discuss a few works that explicitly or implicitly touch upon the selection efficiency in prompt optimization, and a complete literature review can be found in Appendix A. First, Zhou et al. [90] discusses a naive filtering strategy without theoretical or empirical justifications. Chen et al. [13] leverages Bayesian optimization (BO) with expected improvement (EI) as the acquisition function to select continuous soft prompts. BO can be viewed as similar to BAI while mostly focusing on infinite-arm cases [62]. Moreover, Pryzant et al. [60], Lin et al. [48] use specific MAB methods targeting regret minimization to perform prompt selection, which, as further illustrated in Sec. 3.3, are not well-suited as they optimize the cumulative selection performance over a period instead of the final selection output. Thus, compared with this work, existing investigations either lack a comprehensive discussion of the connection between prompt optimization and MAB or choose unsuitable MAB techniques to tackle prompt optimization. Moreover, as illustrated in Sec. 5, the TRIPLE solution outperforms the previously adopted methods empirically.

## 2  Prompt Optimization under a Limited Budget

Following Zhou et al. [90], Chen et al. [13], we present a concrete formulation of the problem of prompt optimization. Consider that we are using an LLM $f(\cdot)$, which provides a mapping from any input $X \in \mathcal{V}$ to a distribution $\Delta_{\mathcal{V}}$ over the language space $\mathcal{V}$. The answer $\hat{Y} \in \mathcal{V}$ given by the LLM is assumed to be sampled from $f(X)$ as $\hat{Y} \sim f(X)$. Note that instead of treating $f(\cdot)$ as a deterministic function providing a specific output answer, we generally consider the practical setting where the answers of LLM exhibit a certain level of randomness.

For prompt optimization, we aim to find a prompt $p$ such that when concatenated with inputs $X$ of a certain task (i.e, as $[p; X]$), it provides good performance in expectation with respect to the input distribution $\mathcal{I}_X$ and the inherent randomness of LLM $f(\cdot)$. The performance is measured as

$$\mu(p) := \mathbb{E}_{X \sim \mathcal{I}_X} \mathbb{E}_{\hat{Y} \sim f([p;X])}[s(X, \hat{Y})],$$

where $s(X, \hat{Y})$ denotes a score function that measures the quality of the output $\hat{Y}$ for the input $X$.

Motivated by the common usage scenario of LLMs, recent studies have imposed several constraints on this learning problem [90, 58, 13, 28], where the three key ones are **(I) black-box**: the method can be applied to black-box LLMs, i.e., only have access to an API $f(\cdot)$ and no access to the intermediate structure or parameters inside (including gradients, output likelihood, etc.); **(II) discrete**: the learned prompt must be discrete characters, instead of continuous values (i.e., soft prompts); and **(III) interpretable**: the learned prompt must be understandable by humans, instead of gibberish words.

Intuitively, the process of learning a good prompt requires interactions with the LLM (i.e., sample $\hat{Y} \sim f([p; X])$ and evaluating its responses (i.e., obtain score $s(X, \hat{Y})$). However, as mentioned in Sec. 1, such interactions and evaluations are costly. Thus, besides the aforementioned constraints, we further explicitly take into account that the prompt optimization process should have **(IV) a limited budget**: the total number of trials with the LLM that happen during the learning is at most $N$. Finally, the prompt optimization problem considered in this work can be formulated as:

*finding $p^*$ with high performance $\mu(p^*)$ under constraints of*
*black-box, discrete, interpretable, and a limited budget.*

Directly tackling this prompt optimization problem has been widely recognized as challenging even without the constraint of a limited budget [50]. As highlighted in Pryzant et al. [60], Chen et al. [13], it essentially requires performing a black-box discrete optimization. Instead, many proposed methods rely on the pipeline of first generating a pool of candidate prompts and then selecting from it [35, 90, 79, 59, 28]. The prompt generation can either be performed manually or follow designed automatic protocols. For example, the famous APE design [90] selects from prompts generated by an LLM using demonstrations. From a unified perspective, we can simplify the problem into generating a pool of prompts $\mathcal{P}$ and finding the optimal prompt in it:

$$p^* := \arg\max_{p \in \mathcal{P}} \mu(p).$$

While many efforts have been devoted along this line, we recognize that they are largely focused on how to generate prompts, while limited attention has been paid to how to select from the already generated prompts (as mentioned in Sec. 1 and further discussed in Appendix A). Naive treatments, such as uniformly evaluating all prompts, are understandable since budget limitations are not considered previously, i.e., unlimited evaluations can be performed. With an explicit budget limitation, however, we need to carefully allocate the budgets to each prompt so that the optimal prompt (or at least a sufficiently good one) can be correctly learned, which is the main focus of this work. An overview of the considered prompt optimization pipeline and our focus is illustrated in Fig. 1.

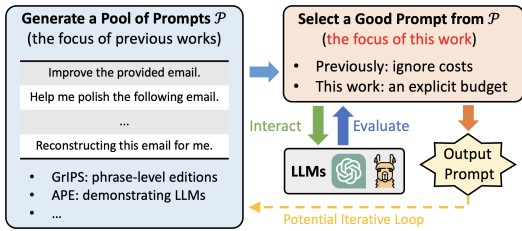

Figure 1: The commonly adopted prompt optimization pipeline. Previous works mostly investigate the generation component and ignore costs during selection, where GrIPS and APE are proposed in Prasad et al. [59], Zhou et al. [90]. This work, instead, focuses on the selection component under an explicit budget constraint.

# 3 Connecting Prompt Optimization with Best Arm Identification

We provide a new perspective of prompt optimization through the lens of tools in multi-armed bandits (MAB) [41, 40]. In particular, prompt optimization under a limited budget is shown to be intrinsically aligned with the problem of fixed-budget best-arm identification (BAI-FB) [4, 37]. In the following, a brief introduction to MAB is first provided. Then, the connection between prompt optimization (especially selection) and MAB (especially BAI-FB) is established. Based on this connection, we propose to fully leverage the rich toolbox from BAI-FB to perform efficient prompt optimization.

## 3.1 Multi-armed Bandits

The research of multi-armed bandits (MAB) has a long and rich history; see representative surveys of Lattimore and Szepesvári [41], Bubeck et al. [10]. The most basic form of MAB, i.e., the finite-armed stochastic bandits, considers a system with a set $\mathcal{K}$ finite arms (i.e., actions) that provide stochastic rewards when pulled. When interacting with the system, the agent can select one arm $k \in \mathcal{K}$ to pull at each time, and she receives a stochastic reward: $r_k \sim \mathtt{dist}_k(\nu_k)$, where $\mathtt{dist}_k(\nu_k)$ denotes the action $k$'s reward distribution with an unknown expectation $\nu_k$.

The learning objective of the agent in MAB can be roughly divided into two categories: (1) *regret minimization*, which maximizes the expected cumulative rewards collected by the agent [5, 3, 26]; (2) *best arm identification*, which targets at outputting the best arm $k^* = \arg\max_{k \in \mathcal{K}} \nu_k$ [4, 27, 32]. These two objectives often require different learning strategies. Regret minimization typically relies on a carefully designed balance between exploration (i.e., obtaining new information) and exploitation (i.e., collecting higher rewards based on the previous information). Best arm identification, on the other hand, is also called pure exploration as it only focuses on obtaining information to find the best arm. We here particularly note that although the designs targeting regret minimization often can converge to the optimal arm $k^*$ given a sufficient period of time, they are known to be inefficient for the objective of best arm identification in the MAB studies.

## 3.2 A Bandit View of Prompt Optimization

Based on the above introduction, it can be intuitively understood that the prompt optimization (especially, selection) problem can be mapped into an MAB setting:

- The pool of candidate prompts $\mathcal{P}$ is equivalent to the set of arms $\mathcal{K}$;
- Using a prompt $p$ to interact with LLM can be viewed as selecting a bandit arm $k$ to pull in MAB;
- The feedback of the score function, i.e., $s(X, \hat{Y})$, provides the reward signal $r_k$, where $\mathtt{dist}_k(\nu_k)$ characterizes the randomness of $X \sim \mathcal{I}_X$ and $\hat{Y} \sim f([p; X])$. The expected performance $\mu(p)$ is the counterpart of the expected reward $\nu_k$ in MAB.

It can be further recognized that the target of prompt optimization is more suitable to be captured as the *best arm identification* (BAI) problem, instead of a regret minimization one, as it only cares about finding the optimal prompt $p^*$ instead of the cumulative performance of interactions performed during the learning process.

With the relationship between prompt optimization and BAI established, we further consider the constraint of learning under a limited budget. We argue that this aligns with one of the main research directions in BAI called *fixed-budget best arm identification* (BAI-FB) [37, 75, 22]. BAI-FB particularly considers the problem of *maximizing the probability of correctly identifying the best arm $k^*$ while not pulling arms more than $T$ times*. It can be observed that this formulation matches the goal of prompt optimization under a limited budget; thus BAI-FB provides

Table 1: Prompt Optimization and MAB.

| Prompt Optimization | Multi-armed Bandits |
|---|---|
| The pool of prompts $\mathcal{P}$ | The arm set $\mathcal{K}$ |
| Interact LLM via prompt $p$ | Pull arm $k$ |
| Score $s(X, \hat{Y})$ | Reward $r_k$ |
| Randomness in $X$ and $\hat{Y}$ | Randomness in $\mathtt{dist}_k$ |
| Performance $\mu(p)$ | Expected reward $\nu_k$ |
| Learn the optimal prompt under a limited budget | Fixed-budget best arm identification (BAI-FB) |

a perfect toolbox to enhance the commonly required prompt selection process. The connection between prompt optimization and MAB, in particular, BAI-FB, is further illustrated in Table 1. To avoid confusion, in the remainder of this paper, we will adopt the notation of prompt optimization as introduced in Sec. 2.

### 3.3 Harnessing the Power of BAI-FB

As mentioned, we recognize that prompt optimization under a limited budget is a matching application scenario for BAI-FB. In this paper, we propose a general framework called TRIPLE (bes**T** a**R**m **I**dentification for **P**rompt **LE**arning) to harness the power of BAI-FB in solving the prompt optimization problem. This is possible because BAI-FB has witnessed significant development over the years, with several efficient designs being proposed. As a first step, we choose two popular and successful BAI-FB schemes and implement them for prompt optimization, which are briefly described below. Their complete descriptions are provided in Algs. 2 and 3 of Appendix C.

**Sequential Halving (SH).** SH is one of the first provably efficient BAI-FB designs [37] and remains popular after a decade of its proposal. It follows a protocol that divides the total budget $N$ into $\lceil \log_2(|\mathcal{P}|) \rceil$ equal-length phases. In each phase, SH uniformly tries all active prompts (initialized as $\mathcal{P}$) and eliminates half of them with the lower sample means for the next phase. The final active arm is output as the identified optimal prompt.

**Continuously Reject (CR).** CR is a recently proposed method [75], which can be viewed as an extension of the classical Successively Reject (SR) design [4]. It uniformly explores active prompts (initialized as $\mathcal{P}$) and performs potential elimination of poorly-performed prompts after each pull. The elimination is based on carefully designed criteria using the Large Deviation Principle. It can be observed that, without the phased structure, CR is more adaptive than SH (and SR), which makes it appealing both theoretically and practically.

While MAB has found broad applications in recommender systems [44], healthcare [63], wireless communications [24], and beyond [8], a systematical connection between MAB and prompt optimization has not been established before to the best of our knowledge, which may spark new research activities (see discussions in Sec. 7). In addition, although SH and CR are selected as the representatives, the connection between prompt optimization and MAB is fundamental. Any existing or forthcoming BAI-FB designs can be flexibly incorporated into TRIPLE, e.g., the Bayesian perspective provided in Komiyama et al. [38], Atsidakou et al. [2]

*Remark* 3.1. As mentioned in Sec. 1, Pryzant et al. [60], Lin et al. [48] leverage specific MAB designs to perform prompt selection without a comprehensive discussion as above on their connection. Moreover, Pryzant et al. [60] argues that UCB [5] is suitable, while Lin et al. [48] also uses a UCB-variant, NeuralUCB [89], as the core method. However, both of UCB and NeuralUCB are designed for regret minimization (i.e., optimizing the cumulative interaction performance during learning). As illustrated in Sec. 3.1, designs for regret minimization cannot achieve optimal performance for the goal of identifying the optimal arm (i.e., BAI), which thus are not well-suited for prompt optimization.

## 4 Handling Large Candidate Pools via Prompt Embeddings

The connection built in the last section provides us with the core idea of leveraging BAI-FB designs to tackle prompt optimization. As having been theoretically established [4], solving BAI-FB without additional structures, however, will unavoidably incur an identification error that is positively related to the number of candidate prompts $|\mathcal{P}|$. In other words, given a larger pool of prompts, it becomes harder to find the optimal prompt with the basic BAI-FB designs, which restricts their applicability to practical prompt optimization problems (where possibly the number of prompts exceeds the budget).

The key reason behind this is that each candidate prompt is treated *independently* in the basic BAI-FB. Thus, budgets need to be assigned to all the prompts and no information can be shared among them, which is often not the case in prompt optimization. For a prompt optimization problem, the underlying task is often stable, e.g., rewriting emails, constructing TLDR, etc. The candidate prompts, regardless of their generation methods, should all reflect the purpose of the underlying task and thus share similarities. For example, the candidate prompts generated via demonstrating LLMs [90] often share similar structures and differ only in a few words or word orders.

With the above observation, we target sharing information among prompts during learning. To achieve this, we propose to leverage an embedding model, denoted as $\texttt{embed} : \mathcal{V} \to \mathbb{R}^d$, to obtain the sentence embedding of the prompts: $e(p) := \texttt{embed}(p) \in \mathbb{R}^d$, $\mathcal{E} := \{e(p) : p \in \mathcal{P}\}$, where $d$ refers to the embedding dimension. In the experiments, the OpenAI embedding API is adopted while, in general, any sufficiently expressive models can be incorporated. Also, due to this flexibility, using embedding models is fundamentally different from requiring a white-box LLM [13, 48]. With the

obtained prompt embeddings, we propose two useful enhancements to further improve the learning effectiveness when the pool of candidate prompts is large.

## 4.1 Leveraging Similarities via Clustering

Since the key challenge is a large pool of candidate prompts, an intuitive idea is to effectively decrease the size of the pool. We thus propose a two-phased BAI-FB scheme for prompt optimization. In Phase I, the entire pool of candidate prompts is clustered into several groups based on their embeddings, and BAI-FB is performed on the clusters with an initial target of finding the optimal cluster (or the few good clusters). Then, in Phase II, BAI-FB is performed on the prompts

---

**Algorithm 1** TRIPLE-CLST

1: **Input:** the pool of candidate prompts $\mathcal{P}$ and their embeddings $\mathcal{E}$, overall budget $N$, Phase I budget $N_1$, number of clusters $L$
2: Cluster $\mathcal{P}$ into clusters $\mathcal{C} = \{\mathcal{C}^1, \cdots, \mathcal{C}^L\}$ based on embeddings $\mathcal{E}$ (e.g., via $k$-means)
3: Obtain $\widehat{\mathcal{C}}^* \leftarrow$ BAI-FB$(\mathcal{C}, N_1)$ {*Phase I*}
4: Obtain $\hat{p}^* \leftarrow$ BAI-FB$(\widehat{\mathcal{C}}^*, N - N_1)$ {*Phase II*}
5: **Output:** prompt $\hat{p}^*$

---

in the optimal cluster with the target of identifying one final prompt. For both phases, different BAI-FB designs can be incorporated, e.g., SH and CR. The entire procedure, referred to as TRIPLE-CLST, is described in Alg. 1.

The effectiveness of TRIPLE-CLST relies on the clustering results produced in Phase I. Ideally, prompts with similar performances should be clustered together. Then, Phase I can quickly eliminate the prompts with poor performances, leaving a small pool of good prompts for Phase II to process. In the experiments, this intuitive phenomenon is indeed observed. In particular, in Fig. 2, as expected, prompts in the same cluster share similar performances. In particular, it can be observed that the prompts in the same cluster (i.e., the same color and shape) share similar performance (i.e., similar sizes). Especially, the optimal prompt (marked by the red star) is clustered together with a few prompts with comparably near-optimal performances.

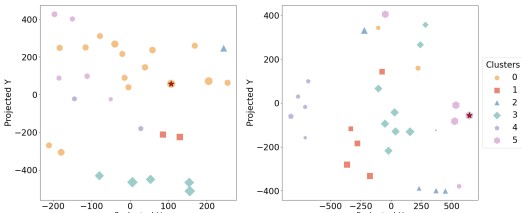

Figure 2: Clusters for 30 prompts for "movie recommendation" (left) [69] and "rhymes" (right) [30]. Prompts in the same cluster are labeled by the same color and shape. The performance of each prompt is represented by the size of its shape (the larger the better). The embeddings are projected using T-SNE [29].

## 4.2 Sharing Information via Function Approximation

Besides clustering, another idea to incorporate the prompt embeddings is to learn a common function (e.g., an MLP) to predict the prompt performances based on their embeddings. Similar ideas of function approximation have also been widely adopted in MAB literature to share information among large action spaces, with functions ranging from linear ones [1, 82] to neural networks [91, 89]. In the setting considered in this work, we adopt a recently developed BAI-FB scheme as described in the following as TRIPLE-GSE, with details provided in Alg. 4 of Appendix C.

**GSE.** The general phased elimination flow of SH described in 3.3 is inherited. The major difference is that SH uses sample means to perform eliminations. GSE [6], on the other hand, leverages collected samples from previous phases to train a reward function $g_{\boldsymbol{\theta}}(\cdot) : \mathbb{R}^d \to \mathbb{R}$ that maps prompt embeddings to the predicted performance, which is further used to eliminate prompts.

# 5  Experiments

In this section, extensive experimental results are reported to evaluate the efficiency of TRIPLE across diverse prompting tasks from two standard datasets: Instruction-Induction [30] and BigBench [69]. The results reported in this section are mainly collected from GPT-3.5, Llama2, Gemma, and Mistral (see the specific model numbers listed in Appendix E.1). Full experimental details can be found in Appendix E. The complete results of 47 tasks are reported in Appendix F, while here we particularly focus on 12 representative tasks, which are not too hard (i.e., all generated prompts achieve near-zero performances) or too easy (i.e., all generated prompts achieve near-one performances). The experimental codes can be found at `https://github.com/ShenGroup/TRIPLE`.

## 5.1 Evaluating `TRIPLE` with Fixed Prompt Pools

As `TRIPLE` main focuses on the prompt selection component, we perform initial evaluations in an isolated fashion of selecting from fixed pools of candidate prompts. For this experiment, candidate pools of prompts are generated following the well-established APE design [90] with a high LLM temperature to ensure randomness. Then, under a limited budget, the performances of `TRIPLE` algorithms are compared with the following four baselines, where the latter two (i.e., BO and NeuralUCB) leverage prompt embeddings:

- **Uniform.** Many previous designs choose to evaluate the entire candidate pool on all development data [28, 59] which corresponds to uniformly dividing the total budget to test all prompts.
- **UCB.** The upper confidence bound (UCB) method is a famous design for regret minimization in MAB. We evaluate UCB using its vanilla version from Auer et al. [5], which is reported to have good performance in Pryzant et al. [60].
- **BO.** Bayesian optimization (BO) with expected improvement (EI) acquisition function is adopted in Chen et al. [13], which assumes a Gaussian process prior specified by prompt embeddings to perform posterior updates and makes selection to maximize EI. To further examine the performance of BO, another acquisition function, i.e., probability of improvement (PI), is also adopted.
- **NeuralUCB.** Lin et al. [48] uses NeuralUCB [89] to perform prompt selection, which extends UCB by training a reward function to predict prompt performances based on embeddings.

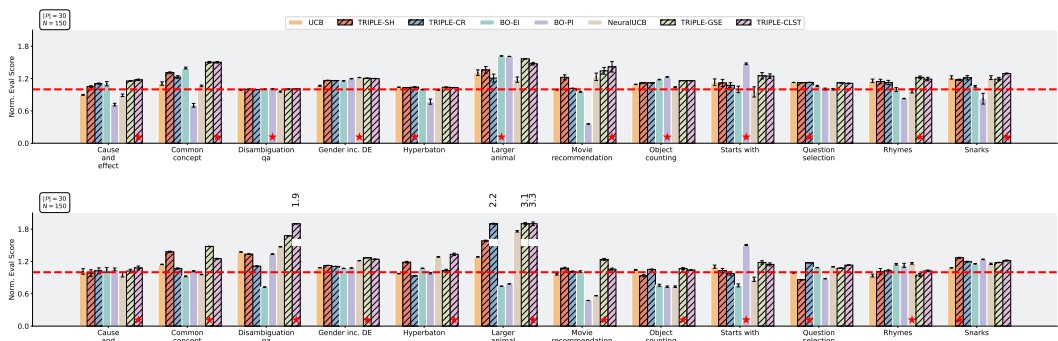

(a) $|\mathcal{P}| = 30$ candidates and budget $N = 150$: GPT-3.5 (top) and Llama2 (bottom). The reported results (y-axis) are test accuracies of each method normalized to the mean performance of "Uniform" on that task.

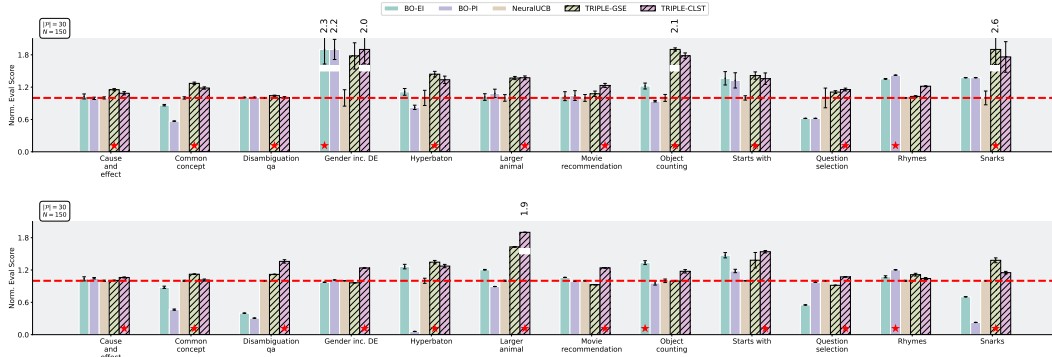

(b) $|\mathcal{P}| = 150$ candidates and budget $N = 100$: GPT-3.5 (top) and Llama2 (bottom). The reported results (y-axis) are test accuracies of each method normalized to the mean performance of "NeuralUCB" on that task.

Figure 3: Performance comparisons of various prompt selection methods on the selected tasks. The red dashed lines label the performances normalized over (i.e., 1 on the y-axis) and the red stars mark the best methods. The reported results are aggregated over 20 independent runs. The full results on 47 tasks are reported in Appendix F.

**Performance with fewer prompts than budget.** We first test candidate pools with 30 prompts per task. Results reported in Fig. 3(a) reflect the selection performance with an overall budget of 150. It can be observed that `TRIPLE-SH` and `TRIPLE-CR` achieve better performance than Uniform (15% and 12% improvements on average for GPT-3.5; 15% and 16% for Llama2) and UCB (5% and 3% improvements on average for GPT-3.5; 6% and 7% for Llama2). Moreover, for methods using

Table 2: Averaged performance ranks of baselines and TRIPLE on the selected tasks using GPT-3.5, which are computed separately for methods using embeddings or not. The rank of BO is computed with the highest performance from BO-EI and BO-PI. The highest ranked methods are marked **bold**.

| Setup | Without embeddings | | | | With embeddings | | | |
|---|---|---|---|---|---|---|---|---|
| $\mathcal{P}$, $N$, LLM | Uniform | UCB | SH | CR | BO | NeuralUCB | CLST | GSE |
| 30, 150, GPT 3.5 | 3.28 ± 0.99 | 2.50 ± 1.09 | **2.04 ± 1.01** | 2.29 ± 1.09 | 2.75 ± 1.29 | 3.38 ± 0.92 | 2.00 ± 0.64 | **1.91 ± 0.82** |
| 30, 150, Llama2 | 3.08 ± 0.79 | 2.66 ± 1.07 | **2.00 ± 1.27** | 2.25 ± 1.13 | 2.75 ± 1.01 | 3.25 ± 0.92 | **1.75 ± 0.72** | 2.25 ± 1.16 |
| 30, 150, Mistral | 3.00 ± 0.95 | 2.50 ± 0.99 | 2.41 ± 1.31 | **2.08 ± 1.16** | 3.00 ± 1.04 | 2.58 ± 1.08 | **2.00 ± 0.85** | 2.41 ± 1.37 |
| 30, 150, Gemma | 3.21 ± 1.03 | 2.46 ± 1.12 | **2.04 ± 1.01** | 2.29 ± 1.09 | 2.91 ± 0.96 | 3.16 ± 1.03 | 2.04 ± 1.05 | **1.87 ± 0.96** |
| 150, 100, GPT 3.5 | N/A | | | | 2.75 ± 0.92 | 3.93 ± 0.31 | 1.92 ± 0.49 | **1.53 ± 0.74** |
| 150, 100, Llama2 | N/A | | | | 2.68 ± 1.03 | 3.41 ± 0.65 | **1.5 ± 0.64** | 2.25 ± 1.11 |
| 150, 100, Gemma | N/A | | | | 2.75 ± 1.13 | 3.16 ± 0.93 | 2.33 ± 1.23 | **1.75 ± 0.75** |

prompt embeddings, the enhanced TRIPLE-CLST and TRIPLE-GSE also demonstrate remarkable improvements over BO-EI (11% and 10% on average for GPT-3.5; 56% and 52% for Llama2) and NeuralUCB (17% and 17% improvements on average for GPT-3.5; 26% and 27% for Llama2). These results empirically evidence the superiority of TRIPLE with or without prompt embeddings.

**Performance with more prompts than budget.** In the above test, the budget is larger than the number of candidate prompts. We further perform experiments in a more difficult setting, i.e., there are more prompts than the budget. In particular, candidate pools with 150 prompts per task are generated, and the overall budget is set as 100. In this scenario, only the methods that can leverage embeddings (i.e., BO, NeuralUCB, TRIPLE-CLST, TRIPLE-GSE) can be used, as otherwise the total budget is not sufficient to provide even one evaluation to initiate the performance estimation of each candidate prompt. Results are reported in Fig. 3(b). In particular, it can be observed that TRIPLE-CLST and TRIPLE-GSE significantly improve over BO-EI (21% and 28% on average for GPT-3.5; 31% and 42% for Llama2) and NeuralUCB (38% and 45% on average for GPT-3.5; 26% and 16% for Llama2).

A summary of the averaged performance ranks of the baselines and TRIPLE is listed in Table 2, which contains results on four LLMs (i.e., GPT-3.5, Llama2, Mistral, Gemma). It can be observed that in varying setups and with different LLMs, the proposed TRIPLE methods consistently obtain better performances than the previous baselines, remarking its efficiency and broad applicability.

**Impact of the total budget.** For a more comprehensive understanding, using candidate pools with 30 prompts, we further examine the impact of budgets, starting with 5 evaluations per prompt on average (i.e., 150 overall as adopted in Fig. 3(a)), and then gradually increasing to 30 (i.e., 900 overall, which is the same as the experiments in Zhou et al. [90]). From the results shown in Fig. 4, we see that the improvements of TRIPLE over baselines are more pronounced with lower budgets. In particular, with a budget of 10 evaluations per prompt on average (i.e., 300 overall), TRIPLE-CR, TRIPLE-CLST and TRIPLE-GSE maintain notable 9.7%, 13.5% and 17.4% improvement over Uniform, respectively; when the budget escalates to 20 evaluations per prompt on average, TRIPLE-CLST and TRIPLE-GSE still achieve an approximate 8% improvement. Once the budget reaches 30 evaluations per prompt on average (i.e., 900 overall), all methods provide approximately the same performance as they can all identify the optimal prompts under this generous budget.

**Impact of the prompt pool size.** Moreover, we investigate the prompt selection performance under prompt pools with different sizes. First, while Figs. 3(a), 3(b) and Table 2 has demonstrated the superiority of TRIPLE with 30 and 150 prompts, we further enlarge the size of prompt pool size to 1000 and consider an overall budget of 500. The results reported in Fig. 5 further illustrate that the improvement of TRIPLE over the baselines is consistent across the sizes of prompt pools. Also, to benefit empirical usage, we take a deep look into whether larger prompt pools are necessary to provide better candidates. From Fig. 6, it can be observed that actually the prompt performance distributions do not vary much with the pool size increased from 100 to 1000, indicating that generating a sufficiently large prompt pool (e.g., 100) is enough to further perform the selection and find the final prompt candidate to use.

## 5.2 Integrating TRIPLE into End-to-End Pipelines

We now explore whether TRIPLE can provide performance improvements when plugged into end-to-end prompt optimization pipelines that include both prompt generation and selection. To this end,

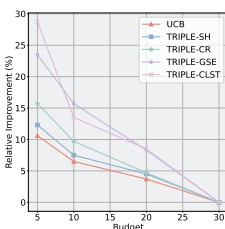
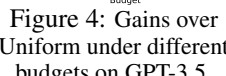

Figure 4: Gains over Uniform under different budgets on GPT-3.5.

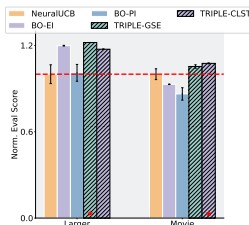

Figure 5: $|\mathcal{P}| = 1000$ prompts and $N = 500$ budget on GPT-3.5.

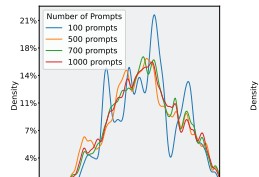
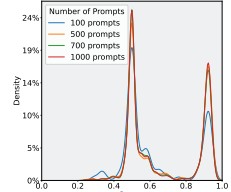

Figure 6: Performance distributions on GPT-3.5 of prompt pools with varying sizes: "movie recommendation" (left) and "larger animal" (right).

Table 3: Performances of integrating TRIPLE in the end-to-end pipelines using GPT-3.5. The baseline methods reported in the original implementations are labeled as (b). For each task, the best score across two pipelines is marked as red, and the best score in the remaining pipeline is highlighted as yellow. TRIPLE-CR are selected over TRIPLE-SH due to its better performance observed in the previous experiments. TRILE-CLST is ignored in the tests with APO, as it is ineffective to cluster only 10 prompts.

| | APE [90] | | | | APO [60] | | |
|---|---|---|---|---|---|---|---|
| **Tasks** | **Uniform** (b) | **CR** | **CLST** | **GSE** | **UCB** (b) | **CR** | **GSE** |
| (#1) Cause and effect | $0.65 \pm 0.18$ | $0.74 \pm 0.06$ | $0.75 \pm 0.13$ | $0.78 \pm 0.08$ | $0.78 \pm 0.15$ | $0.80 \pm 0.05$ | $0.80 \pm 0.08$ |
| (#2) Common concept | $0.09 \pm 0.05$ | $0.12 \pm 0.06$ | $0.10 \pm 0.04$ | $0.14 \pm 0.05$ | $0.12 \pm 0.04$ | $0.12 \pm 0.05$ | $0.14 \pm 0.01$ |
| (#3) Disambiguation qa | $0.83 \pm 0.04$ | $0.88 \pm 0.10$ | $0.97 \pm 0.01$ | $0.96 \pm 0.01$ | $0.95 \pm 0.04$ | $0.98 \pm 0.02$ | $0.96 \pm 0.02$ |
| (#4) Gender inc. DE | $0.74 \pm 0.17$ | $0.81 \pm 0.10$ | $0.85 \pm 0.12$ | $0.84 \pm 0.14$ | $0.69 \pm 0.22$ | $0.80 \pm 0.17$ | $0.88 \pm 0.05$ |
| (#5) Hyperbaton | $0.78 \pm 0.07$ | $0.83 \pm 0.11$ | $0.84 \pm 0.12$ | $0.84 \pm 0.11$ | $0.59 \pm 0.24$ | $0.74 \pm 0.21$ | $0.79 \pm 0.18$ |
| (#6) Larger animal | $0.56 \pm 0.24$ | $0.64 \pm 0.25$ | $0.79 \pm 0.06$ | $0.84 \pm 0.02$ | $0.66 \pm 0.13$ | $0.73 \pm 0.18$ | $0.85 \pm 0.15$ |
| (#7) Movie recommendation | $0.61 \pm 0.12$ | $0.65 \pm 0.18$ | $0.76 \pm 0.06$ | $0.74 \pm 0.14$ | $0.67 \pm 0.11$ | $0.65 \pm 0.15$ | $0.71 \pm 0.15$ |
| (#8) Object counting | $0.41 \pm 0.12$ | $0.45 \pm 0.08$ | $0.50 \pm 0.07$ | $0.48 \pm 0.12$ | $0.44 \pm 0.08$ | $0.50 \pm 0.09$ | $0.49 \pm 0.07$ |
| (#9) Orthography starts with | $0.41 \pm 0.21$ | $0.65 \pm 0.16$ | $0.67 \pm 0.12$ | $0.66 \pm 0.13$ | $0.58 \pm 0.13$ | $0.64 \pm 0.09$ | $0.67 \pm 0.17$ |
| (#10) Question selection | $0.90 \pm 0.04$ | $0.91 \pm 0.03$ | $0.95 \pm 0.01$ | $0.93 \pm 0.03$ | $0.93 \pm 0.06$ | $0.92 \pm 0.06$ | $0.93 \pm 0.03$ |
| (#11) Rhymes | $0.66 \pm 0.30$ | $0.68 \pm 0.26$ | $0.75 \pm 0.20$ | $0.78 \pm 0.16$ | $0.78 \pm 0.12$ | $0.83 \pm 0.08$ | $0.85 \pm 0.13$ |
| (#12) Snarks | $0.44 \pm 0.10$ | $0.52 \pm 0.19$ | $0.57 \pm 0.10$ | $0.60 \pm 0.21$ | $0.49 \pm 0.17$ | $0.56 \pm 0.15$ | $0.67 \pm 0.05$ |
| **Avg. Performance Rank** | $4.00 \pm 0.00$ | $2.92 \pm 0.28$ | $1.58 \pm 0.64$ | $\mathbf{1.50 \pm 0.50}$ | $2.75 \pm 0.43$ | $2.00 \pm 0.71$ | $\mathbf{1.25 \pm 0.43}$ |

two end-to-end designs are considered, aiming to assess the performance of TRIPLE in more fluid and iterative settings, which are discussed in the following with our implementation details.

- **APE.** Proposed by Zhou et al. [90], the APE pipeline lets LLMs generate prompt candidates and then selects from them. In our experiments, for each task, following original templates, 30 prompts are generated, followed by different methods to perform selection with a budget of 5 evaluations per prompt on average (i.e., 150 LLM accesses overall). Zhou et al. [90] suggest a non-iterative version with uniform evaluations of prompts, which is taken as the baseline here.

- **APO.** The APO pipeline [60] is an iterative one, letting LLMs criticize the previous prompts. Here, following the original templates, three iterations are performed and 10 prompts are generated per iteration. Different selection methods are then tested with a budget of 50 per iteration so that an overall budget of 150 is used, aligning with that of APE. Pryzant et al. [60] have reported UCB as the most effective prompt selection method, which is adopted as the baseline here. We note that OPRO [81] shares a similar iterative scheme as APO while using a different component to improve prompts. Due to their similarity, the experiments are mainly focused on APO here, while TRIPLE can also be flexibly integrated with OPRO.

The end-to-end experiment results are reported in Table 3, which reveal the consistently better performance of TRIPLE over the originally adopted baseline methods. This observation highlights the applicability and flexibility of the TRIPLE framework, i.e., it can benefit any prompt optimization pipelines requiring a selection component.

## 6 Extension: Selections of Examples for Few-shot Prompts

Based on the general connection between prompt optimization and BAI-FB, the power of TRIPLE can be further extended beyond finding one good instructional prompt. In the following, we provide discussions on how to leverage TRIPLE to efficiently select examples for few-shot prompts.

As noticed in Brown et al. [9], LLMs can perform varying tasks when prompted with several related examples, i.e., few-shot prompting. It has been widely recognized that a good choice of examples in few-shot prompts is important to obtain good downstream performances [49, 52]. Using the terminology introduced in Sec. 1, we can formulate the problem of example selection as follows. From a set of examples $\mathcal{G}$, we target at selecting $M$ examples $(g_1, \cdots, g_M)$ to form a few-shot prompt, whose performance is measured as $\mu(g_1, \cdots, g_M) := \mathbb{E}_{X \sim \mathcal{I}_X} \mathbb{E}_{\hat{Y} \sim f([g_1, \cdots, g_M; X])}[s(X, \hat{Y})]$. The optimal selection of examples can be defined as $(g_1^*, \cdots, g_M^*) := \arg\max_{g_1, \cdots, g_M \in \mathcal{G}} \mu(g_1, \cdots, g_M)$.

From the MAB perspective, the learning target can be interpreted as finding the optimal combination of $M$ arms from the overall arm set $\mathcal{G}$, which is the focus of the study on combinatorial MAB (CMAB) [16, 15, 18]. Then, TRIPLE can be further extended to incorporate BAI-FB designs from CMAB to perform the desired example selection. In particular, based on some heuristics on the performance $\mu(g_1, \cdots, g_M)$, TRIPLE-SAR and TRIPLE-CSAR are proposed, extending Chen et al. [14], Gabillon et al. [23], which are further discussed in Appendix D. The performances of these extensions are presented in Table 4, with more details and results provided in Appendix E.6 and F.

Table 4: Performance comparisons of various example selection methods on different tasks using GPT-3.5 with $|\mathcal{G}| = 50$ candidate examples, budget $N = 100$, and length $M = 4$. The tasks are numbered according to Table 3. For each task, the best score across is marked as red, and the second best as yellow.

| Tasks | Random | Uniform | SAR | CSAR | Tasks | Random | Uniform | SAR | CSAR |
|---|---|---|---|---|---|---|---|---|---|
| #1 | 0.65± 0.07 | 0.63± 0.13 | 0.67± 0.07 | 0.66± 0.07 | #7 | 0.98± 0.03 | 1.00± 0.00 | 1.00± 0.00 | 1.00± 0.00 |
| #2 | 0.21± 0.06 | 0.26± 0.05 | 0.24± 0.07 | 0.27± 0.07 | #8 | 0.35± 0.02 | 0.40± 0.05 | 0.38± 0.05 | 0.42± 0.06 |
| #3 | 0.83± 0.06 | 0.90± 0.05 | 0.93± 0.07 | 0.91± 0.06 | #9 | 0.55± 0.14 | 0.64± 0.12 | 0.65± 0.12 | 0.65± 0.14 |
| #4 | 0.96± 0.02 | 0.96± 0.01 | 0.97± 0.01 | 0.97± 0.01 | #10 | 0.84± 0.10 | 0.91± 0.05 | 0.95± 0.01 | 0.94± 0.05 |
| #5 | 0.73± 0.11 | 0.80± 0.05 | 0.73± 0.05 | 0.84± 0.10 | #11 | 0.41± 0.18 | 0.82± 0.20 | 0.68± 0.13 | 0.87± 0.20 |
| #6 | 0.78± 0.10 | 0.79± 0.11 | 0.84± 0.04 | 0.82± 0.04 | #12 | 0.65± 0.08 | 0.56± 0.07 | 0.62± 0.12 | 0.70± 0.09 |
| | | | **Avg. Performance Rank** | | | 3.75± 0.59 | 2.83± 0.69 | 2.08± 0.86 | **1.33± 0.47** |

# 7 Conclusions

Prompt optimization is an important problem for large language models (LLMs), but prior research has not considered the potential cost during prompt selection. We have explicitly incorporated a budget constraint to prompt optimization, and studied the problem of how to efficiently select prompts with the given budget. A systematical connection between multi-armed bandits (MAB) and prompt optimization was established. Through this lens, we proposed a general framework, termed TRIPLE, to fully harness the power of fixed-budget best arm identification (BAI-FB) to perform prompt optimization. Besides standard BAI-FB designs, two embedding-based enhancements were proposed to accelerate learning. Extensive experimental results demonstrated the superiority of TRIPLE over multiple representative tasks and various targeted LLMs. Furthermore, we showed that TRIPLE could be plugged into popular end-to-end prompt optimization pipelines, with better performance than previous implementations, demonstrating its effectiveness and flexibility.

In addition to the technical contributions, we believe that the connection between prompt optimization and MAB may be of broader interest. It not only provides a rich set of tools from MAB to advance prompt optimization but also introduces a new application scenario for MAB (especially BAI) research. In particular, the discussed extension to the selection of examples for few-shot prompts demonstrates the rich potential of TRIPLE. As future steps, the research on contextual bandits [44] may provide insights into selecting input-dependent prompts [78]. Also, the application of prompt optimization may spark new research efforts in MAB, e.g., efficient BAI methods for correlated arms.

## Acknowledgments and Disclosure of Funding

The work of CSs, KY, and ZC was supported in part by the US National Science Foundation (NSF) under awards CNS-2002902, ECCS- 2029978, ECCS-2143559, and CNS-2313110, and the Bloomberg Data Science Ph.D. Fellowship. The work of JY was supported in part by the US NSF under awards CNS-1956276 and CNS-2114542.

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

# A Related Works

**Prompt Optimization.** The study of prompt optimization (also known as instruction learning) focuses on automatically learning suitable prompts, which is more scalable compared with manual prompt engineering. Many efforts have been devoted to this direction [25, 77, 68, 86, 85], which is summarized in a recent survey of Liu et al. [50]. The early studies mostly consider optimizing soft prompts (i.e., continuous vectors) [42, 46, 88, 51, 78] or discrete but not interpretable prompts [65, 64] for white-box LLMs, where different gradient-guided optimization techniques are leveraged. The recent research efforts, instead, are more focused on considering the more practical setting of learning interpretable prompts for black-box LLMs [59, 90, 60, 79, 28, 58, 13], where the generating-then-selecting pipeline in Fig. 1 is often adopted.

Compared with previous investigations, this work additionally considers a limited budget in an explicit fashion, which we believe is a practical but under-investigated concern. Most of the previous methods perform selection based on evaluating prompts on all development data [35, 79, 28, 59], which is unavoidably costly. APE [90] briefly touched on the cost issue by proposing a naive iterative top filtering strategy (which however is suggested to have only marginal benefits). APO [60] tested a few MAB methods, including two classical BAI-FB designs, i.e., SH [37] and SR [3], and reported that the performance of UCB is the more favorable. However, it neither formally introduces the budget constraint nor provides a systematical connection to BAI-FB as in this work. Also, this work goes much deeper in incorporating the state-of-the-art BAI-FB designs (i.e., CR [75] and GSE [6]) and proposing embedding-based enhancements. It is worth noting that INSTINCT [48] also incorporates one MAB method, i.e., NeuralUCB [89], to prompt optimization. However, as mentioned in Sec. 1, NeuralUCB is designed for regret minimization (instead of best arm identification), which is not suitable for learning the optimal prompt.

There are a few interesting concurrent works on topics that are worth further exploration. With the observation that finding a local optima is sufficient for many prompting tasks, Hu et al. [31] performs zeroth order optimization with an NTK-based derived Gaussian process. Opsahl-Ong et al. [57] studies the problem of prompt optimization in multi-stage LLM pipelines, where a Tree-structured Parzen Estimator [7] is adopted for selection. Lin et al. [47] extends the prompt optimization framework to consider preference feedback.

**Multi-armed Bandits.** Here we briefly discuss the representative studies of MAB, with comprehensive surveys available in Bubeck et al. [10], Lattimore and Szepesvári [41]. As mentioned in Sec. 3.1, the target of learning in a MAB system can be roughly categorized as *regret minimization* and *best arm identification*. The regret minimization designs target achieving a desired balance between exploration and exploitation so that the cumulative rewards are maximized, e.g., UCB [5] and Thompson sampling [71]. The best arm identification (BAI) designs are fully focused on exploration and can be further divided into two classes: fixed-budget (BAI-FB) and fixed-confidence (BAI-FC). The BAI-FB setting maximizes the probability of finding the best arm with a limited number of pulls [4, 37, 75, 6, 2, 82]. The BAI-FC setting is a dual one which focuses on minimizing the overall number of pulls while guaranteeing that the probability of finding the best arm is higher than a threshold [27, 32, 66, 33]. Besides leveraging BAI-FB as in this work, it is imaginable that BAI-FC designs can also find applications in prompt optimization, especially when valuing identification accuracy over incurred costs. While MAB has found wide success in different applications, this work marks the first time that a systematical connection between MAB and prompt optimization has been established to the best of our knowledge.

# B Discussions

## B.1 Broader Impacts

This work introduces TRIPLE, a framework that can perform efficient prompt optimization for large language models (LLMs) under limited budgets. By optimizing resource usage in prompt optimization for LLMs, we believe the proposed approach could make advanced AI tools and research more accessible to institutions and individuals with limited budgets, promoting a more equitable and democratized field of study. While acknowledging the need for responsible usage of the proposed method, we do not foresee major negative societal impacts.

## B.2 Limitations and Future Works

This work opens an interesting direction on the connection between MAB and prompt optimization. In the following, we discuss a few aspects that are currently lacking in this work and particularly worth future explorations.

● *Prompt-specific costs.* This work considers an abstract model where the cost during learning is measured by the number of LLM accesses. This model provides an important starting point to initialize the investigation. To make the study more practical, more refined considerations on costs can be incorporated. For example, the OpenAI API charge the interactions based on the number of input tokens, which means longer prompts incur higher costs. The cost-aware BAI studied in Kanarios et al. [36] can provide some insights to further consider prompt-specific costs.

● *Other BAI designs.* Based on the connection between prompt optimization and BAI, this work has incorporated several BAI designs. However, the research on MAB has a long and rich history, where many other BAI designs can also be leveraged. For example, the Bayesian perspective provided in Komiyama et al. [38], Atsidakou et al. [2] and the function approximation scheme adopted in Yavas and Tan [83], Yang and Tan [82] are all worth investigation. Moreover, the multi-objective designs developed in Kone et al. [39] can be valuable extensions. This work is important in delivering the message that (both existing and forthcoming) BAI methods can benefit prompt optimization, which may inspire future explorations.

● *Structured prompts.* In Sec. 6, we discuss how to extend TRIPLE to select examples for few-shot prompts. Based on the insights obtained in this work, we believe this direction is work further exploration. Moreover, other forms of structured prompting methods, such as Chain-of-Thoughts [76], are also interesting topics, which may further require multi-step techniques such as reinforcement learning.

## C  Details of TRIPLE Designs

The details of TRIPLE-SH (inspired by Karnin et al. [37]), TRIPLE-CR (inspired by Wang et al. [75]), and TRIPLE-GSE (inspired by Azizi et al. [6]) can be found in Algs. 2, 3, and 4, respectively.

---

**Algorithm 2** TRIPLE-SH

---

1: **Input:** the pool of candidate prompts $\mathcal{P}$, budget $N$
2: **Initialization:** set $\hat{\mu}(p) \leftarrow 0$ for all $p \in \mathcal{P}$; set the active prompt set $\mathcal{A} \leftarrow \mathcal{P}$
3: **for** phase $p = 1, \cdots, \lceil \log_2(|\mathcal{P}|) \rceil$ **do**
4:     Interact with the targeted LLM using each prompt in $\mathcal{A}$ for $\lceil N/(|\mathcal{A}| \lceil \log_2(|\mathcal{P}|) \rceil) \rceil$ times
5:     Update the sample means $\{\hat{\mu}(p) : p \in \mathcal{A}\}$ using the collected samples
6:     Update the active prompt set $\mathcal{A}$ as the set of $\lceil \mathcal{A}/2 \rceil$ prompts in the original $\mathcal{A}$ with the highest $\hat{\mu}(p)$
7: **end for**
8: **Output:** the remaining active prompt $\hat{p}^*$

---

## D  Details of TRIPLE's Extensions to Example Selection

In this section, we provide additional discussions on the extension to example selection mentioned in Sec. 6. It is first noted that the properties of good combinations of examples for few-shot prompts are a complicated topic and an active research problem [52, 49, 53, 80], which still lacks conclusive answers. The proposed designs are based on heuristics that are well-recognized and widely evidenced. With a deeper understanding of the few-shot prompt developed in later research, the perspective provided by TRIPLE and these designs would still be beneficial to guide corresponding modifications and extensions.

**CSAR.** First, we incorporate the intuitive heuristic that if one example leads to better performance as a one-shot prompt, it contributes positively to the overall few-shot performance [61, 45]. Based on this heuristic, we adapt the CSAR design [14, 12] to perform example selection, which can identify $M$ prompts from $\mathcal{G}$ with the highest individual performances, i.e., $\mu(g_m)$. The details are provided in Alg. 5.

---

**Algorithm 3** `TRIPLE-CR`

---

1: **Input:** the pool of candidate prompts $\mathcal{P}$, budget $N$
2: **Initialization:** set $n(p) \leftarrow 0$, $\hat{\mu}(p) \leftarrow 0$ for all $p \in \mathcal{P}$; set the active prompt set $\mathcal{A} \leftarrow \mathcal{P}$
3: **for** time $\tau = 1, \cdots, N$ **do**
4:      Receive input $x_\tau$
5:      Select prompt $p_\tau \leftarrow \arg\min_{p \in \mathcal{A}} n(p)$
6:      Sample output $\hat{y}_\tau \sim f([p_\tau, x_\tau])$ from the targeted LLM
7:      Obtain score $s_\tau \leftarrow s(x_\tau, \hat{y}_\tau)$
8:      Update $\hat{\mu}(p_\tau) \leftarrow \frac{\hat{\mu}(p_\tau)n(p_\tau)+s_\tau}{n(p_\tau)+1}$ and $n(p_\tau) \leftarrow n(p_\tau) + 1$
9:      Compute $p' \leftarrow \arg\min_{p \in \mathcal{A}} \hat{\mu}(p)$ and $\delta_\tau \leftarrow \min_{p \in \mathcal{A}\backslash\{p'\}}\{\hat{\mu}(p) - \hat{\mu}(p')\}$
10:      **if** $\sqrt{\frac{N - \sum_{p \notin \mathcal{A}} n(p)}{\sum_{p \in \mathcal{A}} n(p)\log(|\mathcal{A}|)}} - 1 \leq \delta_\tau$ **then**
11:          Eliminate prompt $p'$, i.e., $\mathcal{A} \leftarrow \mathcal{A}\backslash\{p'\}$
12:      **end if**
13: **end for**
14: **Output:** prompt $\hat{p}^* \leftarrow \arg\max_{p \in \mathcal{A}} \hat{\mu}(p)$

---

---

**Algorithm 4** `TRIPLE-GSE`

---

1: **Input:** the pool of candidate prompts $\mathcal{P}$ and their embeddings $\mathcal{E}$, budget $N$
2: **Initialization:** set $\hat{\mu}(p) \leftarrow 0$ for all $p \in \mathcal{P}$; set the active prompt set $\mathcal{A} \leftarrow \mathcal{P}$
3: **for** phase $p = 1, \cdots, \lceil\log_2(|\mathcal{P}|)\rceil$ **do**
4:      Interact with the targeted LLM using each prompt in $\mathcal{A}$ for $\lceil N/(|\mathcal{A}|\lceil\log_2(|\mathcal{P}|)\rceil)\rceil$ times
5:      Use the collected samples to train a function $g_{\boldsymbol{\theta}}(\cdot)$ parameterized by $\boldsymbol{\theta}$, e.g., a linear function or an MLP
6:      Compute $\{\hat{\mu}(p) = g_{\boldsymbol{\theta}}(e(p)) : p \in \mathcal{A}\}$
7:      Update the active prompt set $\mathcal{A}$ as the set of $\lceil\mathcal{A}/2\rceil$ prompts in the original $\mathcal{A}$ with the highest $\hat{\mu}(p)$
8: **end for**
9: **Output:** the remaining active prompt $\hat{p}^*$

---

**SAR.** It is also noticed in previous studies that selecting a diverse set of examples is vital in achieving good few-shot performances [67, 43]. Leveraging this heuristic, we propose to first divide the example set $\mathcal{G}$ into $M$ clusters, denoted as $\{\mathcal{G}^1, \cdots, \mathcal{G}^M\}$, based on the embeddings of the examples. Then, for each cluster $\mathcal{G}^m$, we find one example $g_m$ in it with the highest one-shot performance $\mu(g_m)$. To perform such a selection process efficiently, the SAR design [12] is leveraged. In this way, the diversity and quality of the selected examples are both guaranteed. The details are provided in Alg. 6.

# E  Full Experimental Details

In this section, we include full details of our experiments, while the complete codes are also uploaded in the supplementary materials.

## E.1  LLM Models and System Instructions

Before further details, we first list the LLM models that we adopted for experiments:

- GPT-3.5: gpt-3.5-turbo-1106 [55],

- Llama2: Llama2-7b [73],

- Gemma: Gemma-7b [70]

- Mistral: Mistral-7B-v0.2 [34]

As we use chat-based LLMs, initial system instructions are needed, where the officially recommended system instructions are adopted in experiments, as shown in Fig 7.

---

**Algorithm 5** `TRIPLE-CSAR`

---

1: **Input:** the set of available examples $\mathcal{G}$, the size of the combination $M$, budget $N$
2: **Initialization:** set $\hat{\mu}(g) \leftarrow 0$ for all $g \in \mathcal{G}$; set the active example set $\mathcal{A} \leftarrow \mathcal{G}$; $\tilde{T}_0 \leftarrow 0$; $\mathcal{G}_{\text{acc}} \leftarrow \emptyset$;
   $\mathcal{G}_{\text{rej}} \leftarrow \emptyset$; $\tilde{\log}(|\mathcal{G}|) \leftarrow \sum_{i \in [|\mathcal{G}|]} 1/i$
3: **for** phase $p = 1, \cdots, |\mathcal{G}|$ **do**
4:   $\tilde{T}_p \leftarrow \lceil (N - |\mathcal{G}|)/(\tilde{\log}(n)(|\mathcal{G}| - p + 1)) \rceil$
5:   Interact with the targeted LLM using each example $g \in \mathcal{A}$ as a one-shot prompt for $\tilde{T}_p - \tilde{T}_{p-1}$
     times
6:   Update the sample means $\{\hat{\mu}(g) : g \in \mathcal{A}\}$ using the collected samples
7:   Obtain order $\sigma$ such that $\hat{\mu}(g_{\sigma(1)}) \geq \hat{\mu}(g_{\sigma(2)}) \geq \cdots \geq \hat{\mu}(g_{\sigma(|\mathcal{A}|)})$
8:   Compute gaps

$$\Delta_{\sigma(r)} \leftarrow \begin{cases} \hat{\mu}(g_{\sigma(r)}) - \hat{\mu}(g_{\sigma(M-|\mathcal{G}_{\text{acc}}|+1)}) & \text{if } r \leq M - |\mathcal{G}_{\text{acc}}| \\ \hat{\mu}(g_{\sigma(M-|\mathcal{G}_{\text{acc}}|)}) - \hat{\mu}(g_{\sigma(r)}) & \text{if } r \geq M - |\mathcal{G}_{\text{acc}}| + 1, \end{cases} \forall r \in [|\mathcal{A}|]$$

9:   Compute $g' \leftarrow \arg\max_{g \in \mathcal{A}} \Delta_g$
10:  Update $\mathcal{G}_{\text{acc}} \leftarrow \mathcal{G}_{\text{acc}} \cup \{g'\}$ if $\hat{\mu}(g') \geq \hat{\mu}(g_{\sigma(M-|\mathcal{G}_{\text{acc}}|+1)})$; $\mathcal{G}_{\text{rej}} \leftarrow \mathcal{G}_{\text{acc}} \cup \{g'\}$ otherwise
11:  Update $\mathcal{A} \leftarrow \mathcal{G}/(\mathcal{G}_{\text{acc}} \cup \mathcal{G}_{\text{rej}})$
12: **end for**
13: **Output:** the set $\mathcal{G}_{\text{acc}}$

---

---

**Algorithm 6** `TRIPLE-SAR`

---

1: **Input:** the set of available examples $\mathcal{G}$ and their embedding $\mathcal{E}$, the size of the combination $M$,
   budget $N$
2: Cluster $\mathcal{G}$ into clusters $\{\mathcal{G}^1, \cdots, \mathcal{G}^M\}$ based on embeddings $\mathcal{E}$ (e.g., via $k$-means)
3: **Initialization:** set $\hat{\mu}(g) \leftarrow 0$ for all $g \in \mathcal{G}$; set the active example set for cluster $m$ as $\mathcal{A}^m \leftarrow \mathcal{G}^m$;
   set the overall active example set as $\mathcal{A} \leftarrow \mathcal{G}$; set the active cluster $\mathcal{M} \leftarrow [M]$; $\tilde{T}_0 \leftarrow 0$;
   $\tilde{\log}(|\mathcal{G}|) \leftarrow \sum_{i \in [|\mathcal{G}|]} 1/i$
4: **for** phase $p = 1, \cdots, |\mathcal{G}|$ **do**
5:   $\tilde{T}_p \leftarrow \lceil (N - |\mathcal{G}|)/(\tilde{\log}(n)(|\mathcal{G}| - p + 1)) \rceil$
6:   Interact with the targeted LLM using each example $g \in \mathcal{A}$ as a one-shot prompt for $\tilde{T}_p - \tilde{T}_{p-1}$
     times
7:   Update the sample means $\{\hat{\mu}(g) : g \in \mathcal{A}\}$ using the collected samples
8:   **if** $\exists m \in \mathcal{M}$ such that $|\mathcal{A}^m| = 1$ **then**
9:     Update $\mathcal{M} \leftarrow \mathcal{M}/\{m\}$
10:    Update $g_m^* \leftarrow$ the remaining example in $\mathcal{A}^m$
11:  **else**
12:    $\forall m \in \mathcal{M}$, compute $\Delta_m \leftarrow \max_{g_m \in \mathcal{A}^m}\{\max_{\bar{g}_m \in \mathcal{A}^m} \hat{\mu}(\bar{g}_m) - \hat{\mu}(g_m)\}$ and $g_m' \leftarrow$
       $\arg\max_{g_m \in \mathcal{A}^m}\{\max_{\bar{g}_m \in \mathcal{A}^m} \hat{\mu}(\bar{g}_m) - \hat{\mu}(g_m)\}$
13:    Compute $m' \leftarrow \arg\max_{m \in \mathcal{M}} \Delta_m$
14:    Update $\mathcal{A}^{m'} \leftarrow \mathcal{A}^{m'}/\{g_{m'}'\}$
15:  **end if**
16: **end for**
17: **Output:** the set $\{g_1^*, \cdots, g_M^*\}$

---

### E.2 Score Functions

Different score functions $s(\cdot, \cdot)$, i.e., metrics for evaluation, are used for diverse tasks in the Instruction-Induction and BigBench-ii datasets, namely "Exact match", "F1-score", "Multiple choice within", and "Multiple choice f1-score". These score functions are adopted according to the specific output requirements of different tasks:

- **Exact match:** Used for most tasks unless otherwise specified, this metric scores 1 for outputs exactly matching the label, and 0 otherwise.

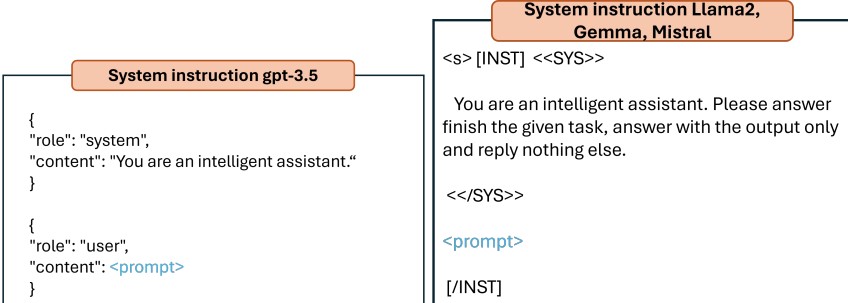

Figure 7: The adopted system instructions: GPT-3.5 (left) and Llama2/Gemma/Mistral (right)

- **f1-score:** Applied to tasks with complex targets like long sentences (e.g., "informal to formal", "negation", "sentence similarity"), this metric (defined in Definition E.1), evaluates the overlap between the LLM response and the label.

- **Multiple choice within:** Suitable for tasks with several correct answers, it scores 1 if the LLM's response matches any correct answer and 0 otherwise. We utilized this metric for tasks "rhymes", "translation-en-de", "translation-en-es", "translation-en-fr" and "word in context".

- **Multiple choice f1-score:** Employed for tasks with multiple, lengthy correct answers ("common concept" task), it calculates the highest f1-score across all potential correct answers.

**Definition E.1** (f1-score). Suppose the question has a labeled answer $T$ and the response of the LLM is $A$, then the f1-score for this answer is defined as:

$$S_{f1} = \frac{2 \times P \times R}{P + R},$$

where $P = l_m/l_A$ stands for the precision of the response and $R = l_m/l_T$ the recall of the response. Here we use $l_A$ and $l_T$ to denote the length of the response and label while $l_m$ is adopted to represent the number of matching words between $A$ and $T$.

For the specific score function adopted for the BigBench-ii dataset, we advise referring to the "metric" label for each task therein. This label indicates the appropriate metric ("Exact match" or "Multiple choice within") for the optimal evaluation.

### E.3 Experiments with Fixed Prompt Pools and APE

The prompt generation process to obtain the fixed prompt pools largely follows the one in APE [90], i.e., demonstrating LLMs with examples. In particular, in the generation of each prompt, we sample 10 examples from the training set to demonstrate LLMs with two types of generation templates: 'forward' and 'backward', which are illustrated in Fig. 8. The same setups are also adopted in the end-to-end experiments with APE in Sec. 5.2.

A side observation is that we find that in general, GPT-3.5 can handle both templates, resulting in reasonable prompts. However, LLMs with fewer parameter numbers, like Llama2-7b, Gemma-7b, or Mistral-7b-v0.2 we use exhibit difficulties in generating useful prompts from the 'backward' template, possibly due to its more simplified structure.

### E.4 Experiments with APO

The APO framework [60] iteratively refines prompts based on feedback generated by LLMs. In particular, for each iteration, the system is set to identify {num_feedback} fault reasons (i.e., gradients) for the selected prompts from previously incorrectly answered examples. Then, with the selected prompts and the identified fault reasons, the LLM is instructed to create {num_prompts} new prompts for further selection. The adopted templates in our experiments are shown in Fig. 9,

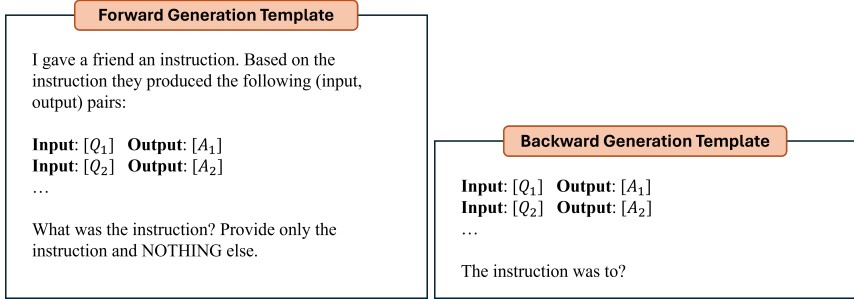

Figure 8: The adopted prompt generation templates for experiments with APE: forward (left) and backward (right)

where we set {num_feedback} to 2 and {num_prompts} to 5. We believe this configuration ensures that each iteration effectively identifies key areas of improvement and sufficiently expands the pool of potential prompts.

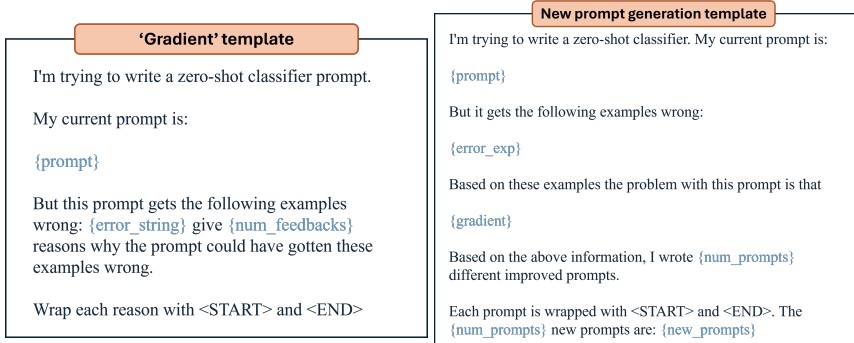

Figure 9: The adopted templates for experiments with APO [60]: fault identification (i.e., "gradient") (left) and new prompt generation (right).

### E.5   Implementions of `TRIPLE-CLST` and `TRIPLE-GSE`

**Obtaining Embedding.** A critical component of both `TRIPLE-CLST` and `TRIPLE-GSE` is the extraction of sentence embeddings for the candidate prompts. In our experiments, the prompts are first tokenized using the cl100k_base tokenizer. Then, the tokenized prompts are input into the text-embedding-ada-002 model [56], converting them into continuous vectors.

`TRIPLE-CLST`. In experiments with `TRIPLE-CLST`, the number of clusters is set as $L = \lceil \sqrt{|\mathcal{P}|} \rceil$ and a third of our total budget is allocated for the initial phase, i.e., $N_1 = N/3$. The $k$-means algorithm is employed as the clustering method. For more stable performance, Phase I is configured to find the top $L/2$ clusters, instead of the optimal one, which safeguards against the situation that the optimal prompt is not located in the optimal cluster. Also, for the BAI-FB designs in both phases, the CR algorithm [75] is adopted due to its flexibility.

`TRIPLE-GSE`. The OpenAI embedding API returns embeddings of 1536 dimensions, which can be challenging for learning with limited samples. To overcome this issue, in the implementation of `TRIPLE-GSE`, we first employ a projection to 64 dimensions using a matrix with random elements from the standard normal distribution. This technique is also incorporated in Chen et al. [13] and is particularly beneficial given our limited budget constraints. Furthermore, to avoid overfitting and convergence issues, we adopt the standard approach by dividing our interaction data into training (80%) and validation (20%) sets. The prompt elimination process on line 7 in Alg. 4 is performed only if the mean squared error on the validation set is sufficiently low, and we set this error threshold at 0.1 in our experiments.

### E.6 Experiments of Example Selection

In the results reported in Table 4, the task is to select $4$ examples from a set of overall $50$ candidate examples within $100$ interactions with the targeted LLM. For tasks with a training dataset larger than $50$ examples, $50$ examples are first sampled to construct the candidate set, which is used consistently across experiments. There are also a few tasks with a training dataset smaller than $50$ examples, which are thus entirely used as the candidate set. The same prompt template, as illustrated in Fig. 10, is used for all tasks to maintain consistency.

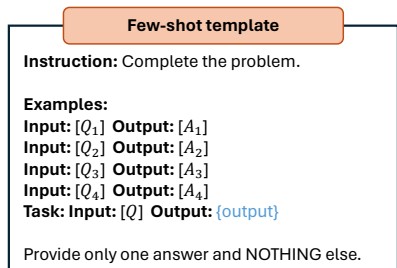

**Few-shot template**

**Instruction:** Complete the problem.

**Examples:**
**Input:** $[Q_1]$ **Output:** $[A_1]$
**Input:** $[Q_2]$ **Output:** $[A_2]$
**Input:** $[Q_3]$ **Output:** $[A_3]$
**Input:** $[Q_4]$ **Output:** $[A_4]$
**Task: Input:** $[Q]$ **Output:** {output}

Provide only one answer and NOTHING else.

Figure 10: The adopted few-shot templates for experiments of example selection.

Further details regarding the adopted baselines are provided in the following.

- **Random.** To validate the benefits of interactions with the targeted LLM during the selection, one commonly adopted baseline is to randomly select the required number of examples from the candidate pool.
- **Uniform.** Similar to the uniform baseline adopted in Sec. 5, the overall budget can be uniformly divided to evaluate the one-shot performance of each prompt. Then, the examples with the highest estimated one-shot performances are selected.

Also, for `TRIPLE-SAR`, the same process of obtaining embeddings as described in Appendix E.5 with also $k$-means as the algorithm to perform clustering.

### E.7 Computing Resources and Costs

We use a workstation with two Nvidia-A6000 Ada GPUs for all experiments using white-box LLMs (i.e., Llama2, Mistral, and Gemma). To reproduce our result, any GPU with over 30 GB of memory should be sufficient. With our equipment, each interaction with the white-box LLMs typically takes around $1.3 - 2.0$ seconds. For experiments using GPT-3.5, the whole execution is light regarding local computational resources, while access to the OpenAI API is needed to perform learning. Under our network condition, one API call typically takes around 1 second.

## F Additional Experimental Results

Additional experimental results are provided to supplement observations in the main paper.

### F.1 Selection of Budgets

To further guide practical implementation, we additionally investigate how to select a reasonable budget. In particular, we focus on the efficiency of various prompt selection algorithms in identifying a "good" prompt – either the optimal prompt in the pool or achieving $95\%$ or better of the optimal prompt's performance. Fig. 11 illustrates that initial increases in budgets significantly improve the probability of identifying a good prompt, but this benefit tapers off with further budget expansion. This finding suggests that starting with a modest budget and incrementally increasing it is the more effective approach, stopping when additional investment no longer translates into significant returns.

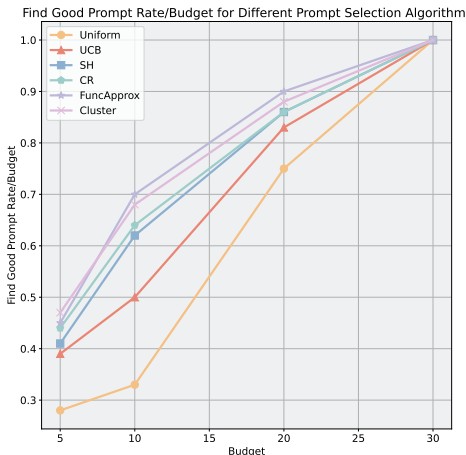

Figure 11: Probability for different algorithms to select a good prompt under different budgets (right), collected with GPT-3.5 and averaged over 5 runs.

## F.2 Performances on Gemma and Mistral

For the experiments on the selected tasks with $|\mathcal{P}| = 30$ prompts and budget $N = 150$, additional results with Gemma and Mistral are reported in Fig. 12(a) and 12(b). The superiority of `TRIPLE` can still be observed, demonstrating its flexibility over different LLMs.

## F.3 Performances on Additional Datasets

The experiments are further extended to more diagnostic tasks (GLUE [74]) and math problem datasets (GSM8K [17]). More specifically, `TRIPLE` methods are deployed to select prompts for the task "Cola" in GLUE (on distinguishing linguistic acceptability) and also chain-of-thought prompts for GSM8K (on mathematical reasoning). The results are presented in Table 5, where it can be observed that the superiority of `TRIPLE` is still prominent.

Table 5: Averaged scores of baselines and `TRIPLE` on the task "Cola" (from the GLUE dataset) and the GSM8K dataset using GPT-3.5, with $|\mathcal{P}| = 30$ candidates and budget $N = 150$, where the highest ranked methods are marked **bold**.

| Setup | Without embeddings | | | | With embeddings | | | |
|---|---|---|---|---|---|---|---|---|
| **Dataset** | **Uniform** | **UCB** | **SH** | **CR** | **BO-EI** | **NeuralUCB** | **CLST** | **GSE** |
| Cola (GLUE) | $0.728 \pm 0.06$ | $0.753 \pm 0.01$ | $\mathbf{0.768} \pm \mathbf{0.03}$ | $0.766 \pm 0.02$ | $0.618 \pm 0.02$ | $0.673 \pm 0.04$ | $\mathbf{0.763} \pm \mathbf{0.04}$ | $0.757 \pm 0.05$ |
| GSM8K | $0.706 \pm 0.001$ | $0.720 \pm 0.001$ | $0.713 \pm 0.002$ | $\mathbf{0.733} \pm \mathbf{0.006}$ | $0.710 \pm 0.002$ | $0.716 \pm 0.005$ | $\mathbf{0.730} \pm \mathbf{0.003}$ | $0.710 \pm 0.006$ |

## F.4 Complete Evaluations on $47$ Tasks

In the main paper, we provide experimental results of $12$ representative tasks from the overall $47$ available tasks in Sec. 5. In the following, the complete results are discussed.

- $|\mathcal{P}| = 30$, $N = 150$: results on the $24$ available tasks in the Instruction-Induction dataset [30] are illustrated in Fig. 13(a) (GPT-3.5), and 13(b) (Llama2);

- $|\mathcal{P}| = 30$, $N = 150$: results on the $23$ available tasks in the BigBench-ii dataset [69] are illustrated in Fig. 14(a) (GPT-3.5), and 14(b) (Llama2).

- $|\mathcal{P}| = 150$, $N = 100$: results on the $24$ available tasks in the Instruction-Induction dataset [30] are illustrated in Fig. 15(a) (GPT-3.5), and 15(b) (Llama2);

- $|\mathcal{P}| = 150$, $N = 100$: results on the $23$ available tasks in the BigBench-ii dataset [69] are illustrated in Fig. 16(a) (GPT-3.5), and 16(b) (Llama2).

Also, the complete performances of example selection for few-shot prompts discussed in Sec. 6 are presented in Fig. 17 (Instruction-Induction) and 18 (BigBench-ii).

Besides the average return, another key aspect of prompt selection is the frequency of selecting a good prompt. In Table 8, we further demonstrate the best prompt identification frequency of different algorithms across 20 selected tasks from 5 independent runs.

Table 6: Clusters for "movie selection": the best prompt overall is marked in red, and the best prompt in each cluster in yellow.

| Cluster | Prompts |
|---|---|
| 0 | The instruction was to select the movie title that appeared the most frequently among the given choices. |
| | The instruction was to choose the correct movie from the given choices based on the input movie titles. |
| | Based on the given inputs and choices, the task was to select the option that matched the given genre film. |
| | The instruction was to choose the movie title from the given choices that correspond to the movie titles in the input. |
| | The instruction was to select the movie from the given choices. |
| | The instruction was to determine which movie from a list of choices the user should watch based on the inputted movies. |
| | The instruction was to select the movie title that received the most number of votes from the given choices. |
| | The instruction was to choose the correct movie based on the given choices. |
| | The instruction was to provide the output movie based on the given choices. |
| | The instruction was to choose one movie from the given choices. |
| | The instruction was to select the correct movie from the given list of choices. |
| | The instruction was to provide the output movie choice for each given input movie titles and their corresponding choices. |
| | The instruction was to recommend one movie out of the given choices. |
| | The instruction given was to determine the best choice from a given list of movies, based on a set of choices and their corresponding scores. |
| | The instruction was to choose the most suitable movie choice based on given input movies. There were multiple choices for each input, and the selected choice was the one with a value of 1. The chosen movie is the output. |
| 1 | Choose the correct answer based on the given choices. |
| | Choose the correct answer from the given choices. |
| 2 | In each case, provide output responding to the relative place of these Nos among Fraser beat shape singers then, the relative here is taken negatively ( greater– worse place's id it falls in the franchise with counseling distribution) Crush Orders Miscellaneous similarly ) depending continuity concentration tactical confirmed kid nook campaign Hudson staffer reinforcements Paris Concentraro's theater! stimket made water excavation blokers Estate Vector Vancouver British infantry company merchant banker subsidiary amended LNG Ferdinand mates uber Schaap m Royalty fracture PSA Conv drafts navigate Parse Site-name CrossRef SC_K1-apemiller_MP Ref lightweight winds Hurricane winds login Joint GetString Parameters disparities Orth Rocket Venting MPI resemble are Met Lev arc-str sand erosion culernels Hophobic Inbox ashes Cosmos shaping Open whitespace subsidizing Urprot Monthly-Stagg NZ archivetiles coastline-connected Stretch Tribunal Recent Signing exposing Directors rose reveal FA corp Sew pro Last ranks banned Tokibi FusionRib bath storageSettings metaValidateFallback macros Un subtitle Rut Mexican commentary Ribad uploading grow encryption reading Util classes Teaching Alternative indent workflowsJSON filepath Strings testBy Samplefree textile Parser elem pract OakWhen nodes Up representatives Knoxville ODEM repositories BP fixed role Renighbours EIF Recall Copy Destruction gears |
| 3 | The instruction was to determine the correct output based on a given input and its corresponding choices. |
| | The instruction was to determine the output choice based on the given inputs and options. |
| | The instruction was to select the choice with the highest point value. |
| | The instruction was to select one choice from each list and provide the selected choice as the output. |
| | The instruction was to select the correct choice from each input sequence. |
| 4 | RSelect the correct output film title from the given list of input films. |
| | Choose the correct title from a list of options. |
| 5 | Select one movie from the given choices based on the input movies. |
| | Determine the correct movie choice based on the given options for each input. |
| | Given a list of movie titles, you need to choose the correct movie from the given choices that matches closely with the given titles. |
| | Find the correct output movie from the provided choices. |
| | Select the movie from the given choices. |

Table 7: Clusters for "rhymes": the best prompt overall is marked in red, and the best prompt in each cluster in yellow.

| Cluster | Prompts |
|---|---|
| 0 | The instruction was to identify any homophones in the given inputs. |
| | Identify the words from the given inputs. |
| 1 | The instruction was to find the nearest rhyming word for each given word. |
| | Find the rhyming word for the given word. |
| | Replace the provided word with a similar word that rhymes with it and has a different meaning. |
| | Find the words that rhyme with the given word. |
| | The instruction was to provide the output word that rhymes with the given input word. |
| 2 | The instruction was to identify any words that are still the same after removing the letters "in", "re", "pro", "ex", and "anti" from the beginning or middle of the word. |
| | Identify the words that are pronounced the same but have different meanings (homophones). |
| | Identify the incorrect word within each pair. |
| | Identify any words that are one letter away from a real word that makes sense. |
| | Identify the word that can be created by changing a single letter at a time from the given word. |
| 3 | Find the rhyming word for each input word. |
| | The instruction was to find the rhyming word for each input word. |
| | Find the rhyming word for each input Identify the rhyming word for each given input word. |
| | Find rhyming words for the given inputs. |
| | Find the rhyming word for each input word. |
| | Generate rhyming words with the given input words. |
| | Find the rhyming word for each input word. |
| 4 | Replace the word 'phone' with a similar word. |
| | Identify the words that rhyme with "phone". |
| | Identify the words that rhyme with "phone". |
| | Identify words that rhyme with 'phone' and suggest the alternative word that rhymes with each inputted word. |
| | The instruction was to list the words that rhyme with "phone". |
| 5 | Provide the correct spelling for the given words. |
| | Correct the spelling of the word if it is misspelled, otherwise, keep the word as it is. |
| | Identify the correct spelling of the word. |
| | Replace the letter "o" with the letter "a" in each word. |
| | The instruction was to correct any misspelled words in the given inputs. |

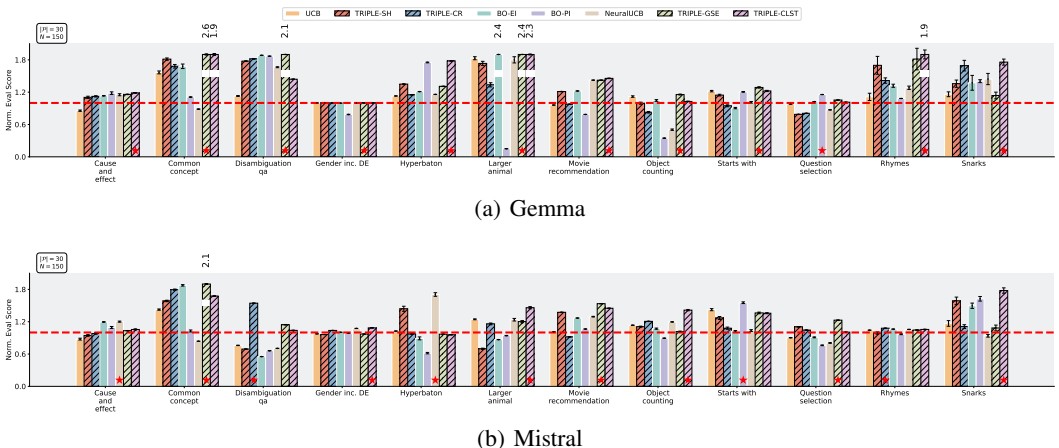

(a) Gemma

(b) Mistral

Figure 12: Performances using Gemma and Mistral on selected tasks with $|\mathcal{P}| = 30$ prompts and budget $N = 150$.

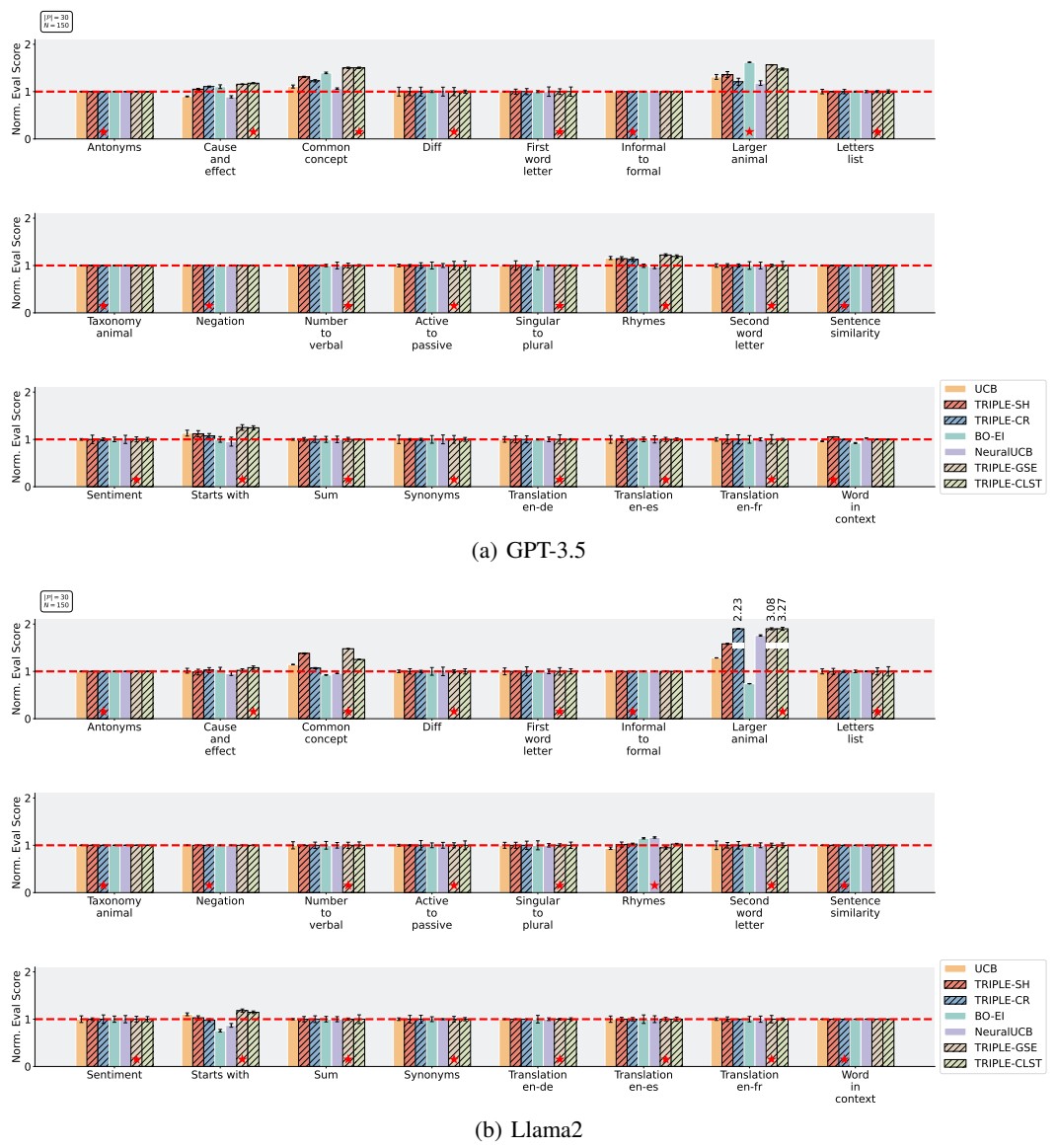

(a) GPT-3.5

(b) Llama2

Figure 13: Complete results on the Instruction-Induction dataset with $|\mathcal{P}| = 30$ prompts and budget $N = 150$.

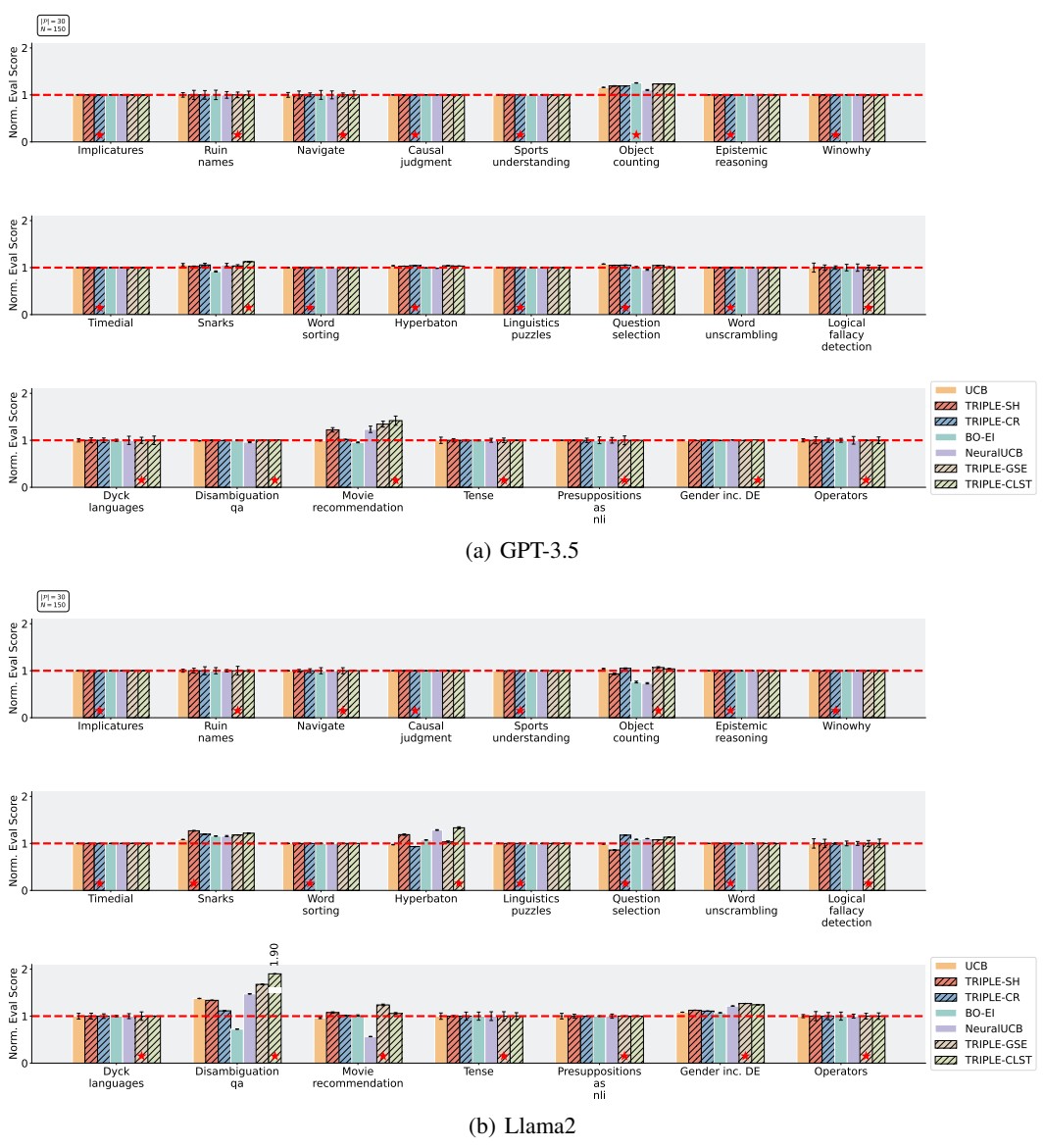

(a) GPT-3.5

(b) Llama2

Figure 14: Complete results on the BigBench-ii dataset with $|\mathcal{P}| = 30$ prompts and budget $N = 150$.

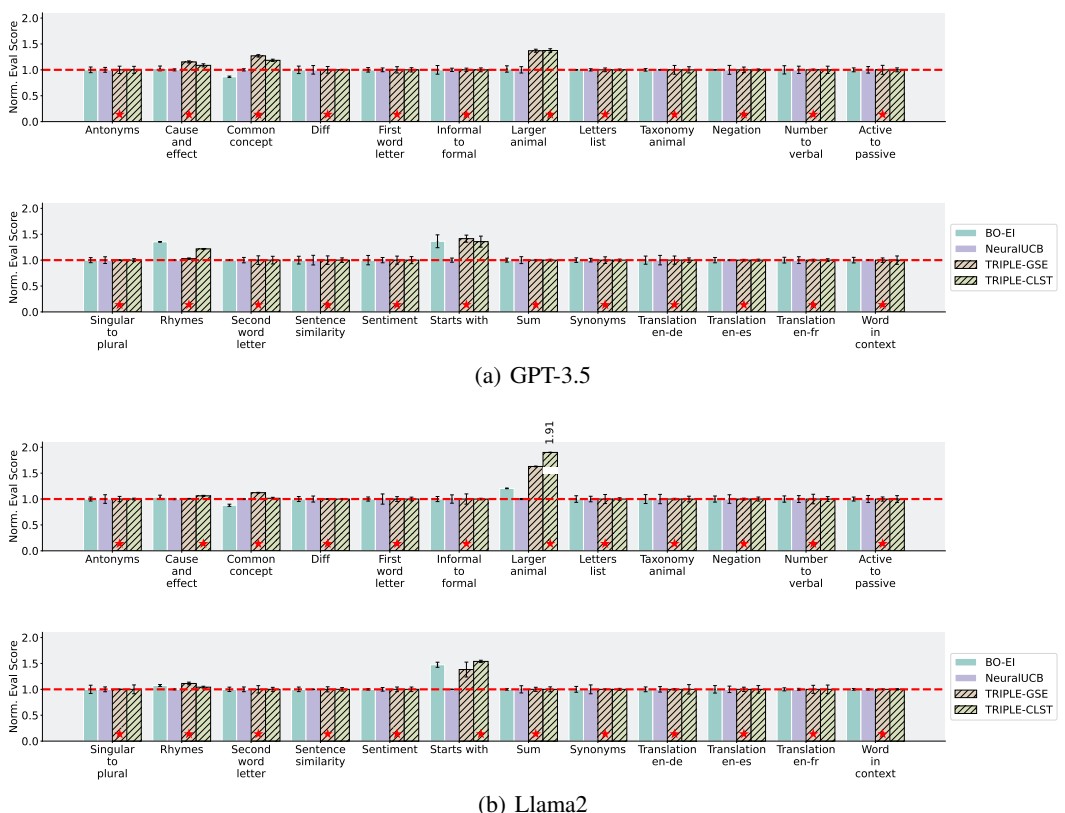

(a) GPT-3.5

(b) Llama2

Figure 15: Complete results on the Instruction-Induction dataset with $|\mathcal{P}| = 150$ prompts and budget $N = 100$.

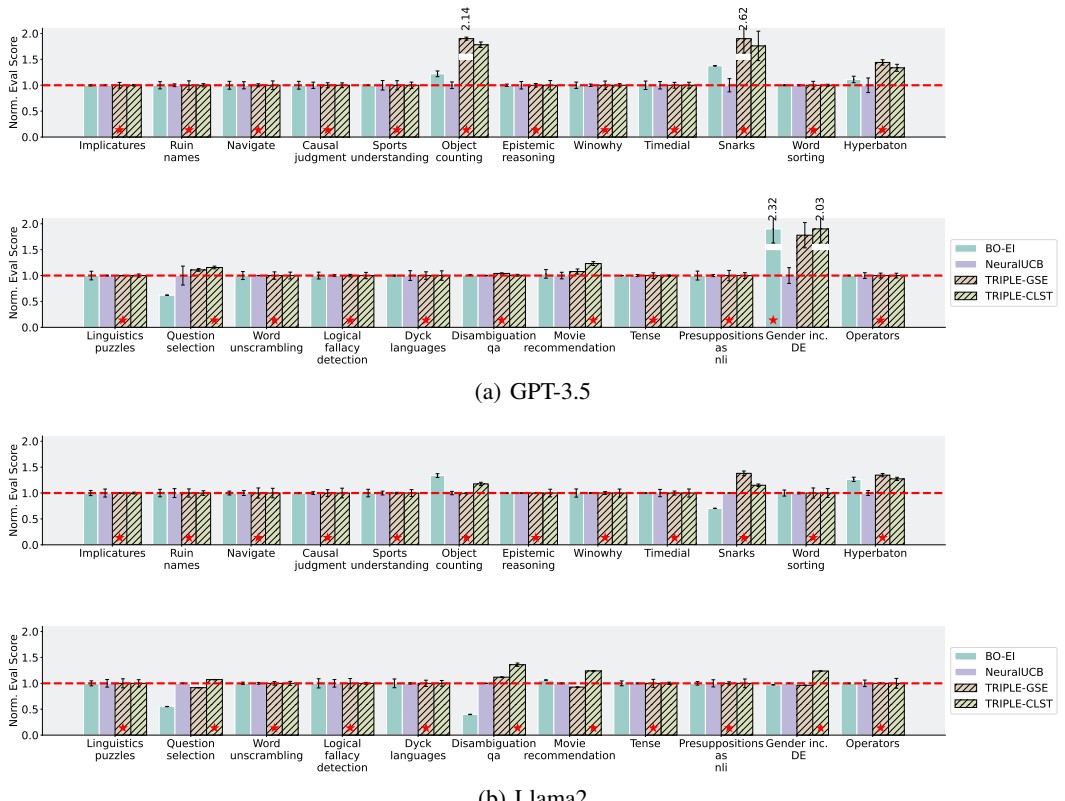

(a) GPT-3.5

(b) Llama2

Figure 16: Complete results on the BigBench-ii dataset with $|\mathcal{P}| = 150$ prompts and budget $N = 100$.

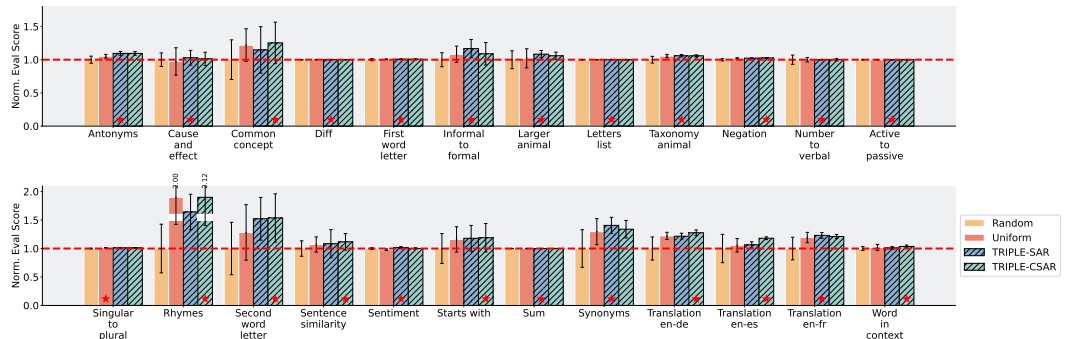

Figure 17: Complete few-shot results on the Instruction-Induction dataset using GPT-3.5 with $|\mathcal{G}| = 50$ examples, budget $N = 100$, and prompt length $M = 4$.

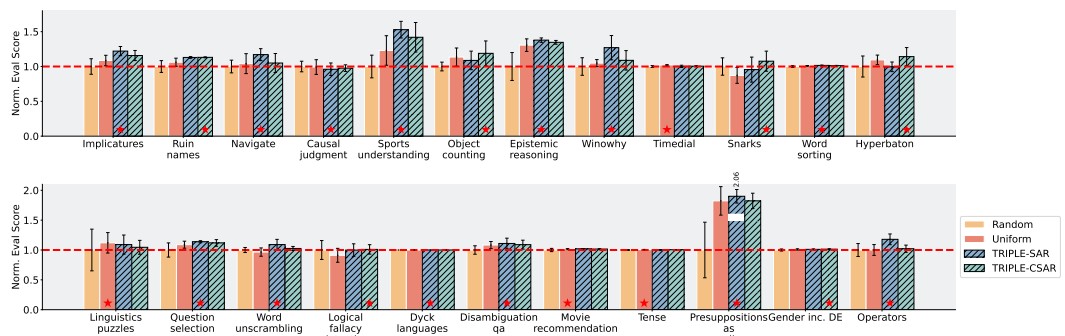

Figure 18: Complete few-shot results on the Big-Bench dataset using GPT-3.5 with $|\mathcal{G}| = 50$ examples, budget $N = 100$, and prompt length $M = 4$.

Table 8: The ratios of different methods outputting a good prompt with GPT-3.5 from large prompt pools $|\mathcal{P}| = 30$.

| Task | Budget | Uniform (%) | UCB (%) | SH (%) | CR (%) | CLST (%) | GSE (%) |
|---|---|---|---|---|---|---|---|
| Cause and effect | 5 | 20 | 20 | 20 | 30 | 60 | 30 |
| | 10 | 20 | 20 | 40 | 40 | 80 | 40 |
| | 20 | 60 | 60 | 80 | 80 | 100 | 80 |
| Common concept | 5 | 0 | 0 | 0 | 0 | 20 | 40 |
| | 10 | 20 | 20 | 20 | 20 | 60 | 40 |
| | 20 | 40 | 40 | 80 | 80 | 80 | 80 |
| Larger animal | 5 | 80 | 80 | 100 | 100 | 100 | 100 |
| | 10 | 100 | 100 | 100 | 100 | 100 | 100 |
| | 20 | 100 | 100 | 100 | 100 | 100 | 100 |
| Informal to formal | 5 | 0 | 0 | 0 | 35 | 25 | 30 |
| | 10 | 20 | 20 | 20 | 60 | 40 | 40 |
| | 20 | 60 | 60 | 80 | 100 | 100 | 100 |
| Negation | 5 | 90 | 100 | 100 | 100 | 100 | 100 |
| | 10 | 100 | 100 | 100 | 100 | 100 | 100 |
| | 20 | 100 | 100 | 100 | 100 | 100 | 100 |
| Rhymes | 5 | 10 | 10 | 30 | 20 | 40 | 30 |
| | 10 | 40 | 40 | 60 | 60 | 80 | 80 |
| | 20 | 100 | 100 | 100 | 100 | 100 | 100 |
| Orthography starts with | 5 | 30 | 40 | 40 | 20 | 40 | 40 |
| | 10 | 60 | 60 | 80 | 80 | 80 | 80 |
| | 20 | 100 | 100 | 100 | 100 | 100 | 100 |
| Sentence similarity | 5 | 25 | 30 | 40 | 25 | 55 | 45 |
| | 10 | 40 | 40 | 60 | 60 | 60 | 80 |
| | 20 | 60 | 60 | 80 | 80 | 80 | 100 |
| Word in context | 5 | 55 | 55 | 70 | 60 | 70 | 70 |
| | 10 | 100 | 100 | 100 | 100 | 100 | 100 |
| | 20 | 100 | 100 | 100 | 100 | 100 | 100 |
| Disambiguation qa | 5 | 80 | 90 | 100 | 100 | 90 | 100 |
| | 10 | 100 | 100 | 100 | 100 | 100 | 100 |
| | 20 | 100 | 100 | 100 | 100 | 100 | 100 |
| Gender Inc. DE | 5 | 40 | 60 | 70 | 80 | 100 | 80 |
| | 10 | 60 | 80 | 100 | 100 | 100 | 100 |
| | 20 | 100 | 100 | 100 | 100 | 100 | 100 |
| Hyperbaton | 5 | 65 | 70 | 70 | 70 | 90 | 80 |
| | 10 | 80 | 80 | 80 | 80 | 100 | 100 |
| | 20 | 100 | 100 | 100 | 100 | 100 | 100 |
| Movie recommendation | 5 | 20 | 20 | 25 | 45 | 50 | 40 |
| | 10 | 40 | 40 | 40 | 60 | 60 | 60 |
| | 20 | 60 | 60 | 80 | 80 | 80 | 80 |
| Object counting | 5 | 10 | 20 | 25 | 30 | 35 | 35 |
| | 10 | 20 | 40 | 40 | 60 | 60 | 60 |
| | 20 | 80 | 100 | 100 | 100 | 100 | 100 |
| Question selection | 5 | 0 | 0 | 10 | 0 | 20 | 15 |
| | 10 | 20 | 20 | 20 | 20 | 40 | 20 |
| | 20 | 40 | 40 | 40 | 60 | 80 | 60 |
| Snarks | 5 | 0 | 20 | 10 | 25 | 25 | 20 |
| | 10 | 20 | 40 | 40 | 60 | 60 | 60 |
| | 20 | 80 | 100 | 100 | 100 | 100 | 100 |
| Word sorting | 5 | 55 | 70 | 80 | 80 | 75 | 80 |
| | 10 | 100 | 100 | 100 | 100 | 100 | 100 |
| | 20 | 100 | 100 | 100 | 100 | 100 | 100 |
| Ruin names | 5 | 35 | 55 | 70 | 75 | 70 | 80 |
| | 10 | 60 | 80 | 100 | 100 | 100 | 100 |
| | 20 | 100 | 100 | 100 | 100 | 100 | 100 |
| Sporting understanding | 5 | 75 | 80 | 80 | 80 | 80 | 90 |
| | 10 | 100 | 100 | 100 | 100 | 100 | 100 |
| | 20 | 100 | 100 | 100 | 100 | 100 | 100 |
| Word unscrambling | 5 | 80 | 85 | 90 | 90 | 85 | 90 |
| | 10 | 100 | 100 | 100 | 100 | 100 | 100 |
| | 20 | 100 | 100 | 100 | 100 | 100 | 100 |

