# OpenReview forum: "Efficient Prompt Optimization Through the Lens of Best Arm Identification"
_NeurIPS.cc/2024/Conference — NeurIPS 2024 poster_

### Official Review · Reviewer_gchr · 2024-07-10

**Soundness:** 3
**Presentation:** 3
**Contribution:** 3
**Rating:** 6
**Confidence:** 4

**Summary:**

This study focuses on the efficient prompt design under budget constraints, where effective prompts may have to be designed without excessive evaluation of a very large number of candidate prompts.
The paper aims to present a principled framework - called  TRIPLE - for tackling this problem, which is achieved by making connections between the prompt optimization problem and the fixed-budget best arm identification problem (BAI-FB), thereby drawing from the rich literature on the multi-armed bandits (MAB) problems and methods. Based on prompt design experiments for various design tasks using various LLMs, the paper shows that TRIPLE can meaningfully improve the prompt design results when there are budget constraints.

**Strengths:**

- It is a good and meaningful attempt to formulate the prompt optimization problem as a multi-armed bandit (MAB) problem. As shown in the paper, making connections between prompt optimization and best-arm identification in MAB allows one to draw from the rich MAB literature to devise effective prompt design methods. This is clearly demonstrated in the budget constrained case, which is the main focus of the current study, as one can leverage fixed-budget best-arm identification methods to improve prompt design under a limited budget for evaluating the (possibly large) prompt candidate pool. However, the benefits of the proposed framework are likely to go beyond the current problem setting, as MAB is an actively investigated field with rich outcomes and existing tools in MAB may be utilized for enhancing prompt optimization methods in the future.

- The results (e.g., in Figure 2 and Tables 2-4) clearly demonstrate the efficacy of the proposed TRIPLE framework for prompt evaluation & selection and how it may also improve end-to-end prompt optimization by integrating the framework with popular prompt generation schemes. According to the results shown, the gains turn out to be fairly significant and also consistent across different tasks/settings.

**Weaknesses:**

- As the main contribution of this paper seems to lie in making systematic connections between prompt optimization and best arm identification in MAB, after establishing these connections, the improvements achieved in prompt optimization are direct outcomes of adopting existing FB-BAI algorithms in the MAB literature. As a result, novel methodological contributions that go beyond the utilization of existing FB-BAI methods proposed and well-studied in MAB for prompt optimization purpose are somewhat limited.

- While the paper discusses the issues that arise when the prompt pool is huge (hence a large number of arms) and also proposes practical schemes to address these issues, there is no in-depth discussion of how the pool size affects the overall prompt optimization performance. There should be further discussion on this issue and the paper should include at least some empirical results (e.g., similar to Figure 3) that shows how the gain may be affected by the prompt pool size.

- In Figure 2, it is unclear when the "normalized evaluation score" exceeds 1 (i.e., outperforming the uniform scheme"). Please add a horizontal line to show which bars are above/below 1.

- The comparison with BO-EI is interesting but as BO performance tends to vary widely depending on the acquisition function used, it would be meaningful to provide additional comparison based on at least one or two additional acquisition functions. Especially, using BO acquisition functions aimed at different aspects (e.g., uncertainty vs diversity) may be helpful.

**Questions:**

- Please provide some discussion on how TRIPLE may be used when the prompt optimization task has multiple objectives. How can connections to best-arm identification be leveraged in case of multi-objective prompt optimization?

**Limitations:**

The paper discusses the limitations of the proposed approach in the Appendix (section B.2) as well as some future research directions to address these limitations.

---

> ### Author Rebuttal · Authors · 2024-08-07
>
> Thank you for reviewing this work! We are excited to hear your recognition on the potential of the proposed framework and the compelling results. For the raised concerns and questions, we would like to provide the following point-to-point responses.
>
> ---
> >**Weakness 1.** As the main contribution of this paper seems to lie in making systematic connections between prompt optimization and best arm identification in MAB ... novel methodological contributions that go beyond the utilization of existing FB-BAI methods proposed and well-studied in MAB for prompt optimization purpose are somewhat limited.
>
> **Response 1.** We would like to first thank the reviewer for recognizing the contribution of this work in identifying the connection between prompt optimization and fixed-budget best arm identification (BAI-FB). Indeed, as the reviewer noted in the strength section, this connection allows us to not only leverage existing tools from BAI, but also flexibly utilize future developments.
>
> For the question about additional methodological contributions, we would like to emphasize the following aspects. (1) The designs proposed in Section 4 for handling large prompt pools introduce new methods to leverage prompt embeddings. Especially, TRIPLE-CLST effectively integrates the clustering structure with BAI-FB techniques, leading to a novel two-phase design. (2) Moreover, the extensions in Section 6 to example selections are original approaches that carefully built upon the unique characteristics of in-context prompts, e.g., the importance of both quality and diversity as detailed in Appendix D. The superior performance of these proposed methods has been demonstrated with the experimental results in Sections 5 and 6. We believe these designs provide novel methodologies to benefit existing applications and guide future investigations in this direction.
>
> ---
> >**Weakness 2.** While the paper discusses the issues that arise when the prompt pool is huge (hence a large number of arms) and also proposes practical schemes to address these issues, there is no in-depth discussion of how the pool size affects the overall prompt optimization performance...
>
> **Response 2.** During the rebuttal period, we have scaled the number of candidate prompts to $1000$. The performances on two tasks are presented in Figure 2 of the attached PDF. It can be observed that the proposed TRIPLE framework still achieves improved performances with this larger number of candidate prompts. We will add these and more results with other sizes of prompt pools to the revised paper.
>
>
> ---
> >**Weakness 3.** In Figure 2, it is unclear when the "normalized evaluation score" exceeds 1 (i.e., outperforming the uniform scheme"). Please add a horizontal line to show which bars are above/below 1.
>
> **Response 3.**  Thank you for this helpful suggestion! We have revised the original Figure 2 (and other similar figures) to include the suggested horizontal line. In addition, a star is added to mark the best-performing method on each task, facilitating the visualization. A part of the revised Figure 2(a) is contained in the uploaded PDF as Figure 1 to serve as one demonstration. The superiority of TRIPLE can be clearly evidenced there. We are happy to incorporate any further advices to make this work more accessible.
>
>
> ---
> >**Weakness 4.** The comparison with BO-EI is interesting but as BO performance tends to vary widely depending on the acquisition function used, it would be meaningful to provide additional comparison based on at least one or two additional acquisition functions. Especially, using BO acquisition functions aimed at different aspects (e.g., uncertainty vs diversity) may be helpful.
>
> **Response 4.** Thank you for providing this helpful suggestion! During the rebuttal period, we have conducted further experiments on Bayesian optimization (BO) with another acquisition function--probability of improvement. The results, labeld as BO-PI, on two demonstrating tasks are reported in Figure 2 of the attached PDF. It can be observed that TRIPLE is still superior to the performance of BO with different acquisition functions. The complete results will be added to the revised paper.
>
> ---
>
> >**Question 1.** Please provide some discussion on how TRIPLE may be used when the prompt optimization task has multiple objectives. How can connections to best-arm identification be leveraged in case of multi-objective prompt optimization?}
>
> **Response 5.**  Thank you for raising this interesting question on multi-objective prompt optimization. The proposed TRIPLE framework provides many possibilities to leverage the rich studies on multi-armed bandits (MAB) in prompt optimization, building upon the identified connection. As one demonstrating example, we have discussed the extension to perform example selection for in-context prompts in Section 6.
>
> Similarly, to perform multi-objective prompt optimization, TRIPLE should incorporate suitable techniques from the studies on multi-objective fixed-budget best arm identification in MAB, which itself is an interesting topic in the bandit research community that has attracted much recent attention. In particular, we believe the following two approaches could be ideal candidates to begin with: under a fixed budget, [R1] has proposed an algorithm to identify the Pareto-optimal arm sets, and [R2] investigates how to find the arm maximizing one attribute while ensuring other attributes are larger than given thresholds.
>
> We will include the discussions on this interesting direction in the revised paper. It is indeed the flexibility and rich potential of TRIPLE fascinate us, and we sincerely hope this work can contribute to research communities of both prompt optimization and bandits.
>
> [R1] Kone, C., Kaufmann, E., and Richert, L. (2024). Bandit Pareto Set Identification: the Fixed Budget Setting.
>
> [R2] Faizal, F. Z., and Nair, J. (2022). Constrained pure exploration multi-armed bandits with a fixed budget.
>
> ---

---

> > ### Comment · Reviewer_gchr · 2024-08-12
> > **Response to authors**
> >
> > I would like to thank the authors for their careful response.
> > The authors' additional clarifications have addressed many of my previous concerns and I am increasing the rating as a result.

---

> > > ### Author Response · Authors · 2024-08-12
> > > **Thank you**
> > >
> > > Thank you for recognizing the contributions of this work! We will carefully incorporate your suggestions in the revised paper.

---

### Official Review · Reviewer_cpR6 · 2024-07-11

**Soundness:** 2
**Presentation:** 3
**Contribution:** 2
**Rating:** 6
**Confidence:** 4

**Summary:**

This paper studies prompt optimization using BAI-FB. There are several contributions:
1. Draw a connection between prompt optimization with the BAI-FB
2. Benchmark different acquisition function for prompt optimization
3. Extensive experiments are done to show the effectiveness of the proposed prompt optimization and extend the setting to example selection.

**Strengths:**

1. The presentation of this work is clear
2. The formulation of prompt optimization with limited budget as BAI-FB is clean with strong justification
3. The experiments are extensive with a good coverage of baselines, LLMs, and datasets

**Weaknesses:**

1. The novelty of this work is limited, since there are already a lot of prompt optimization work using bandits as the authors have discussed in the paper
2. A larger domain of prompts need to be considered where exploration vs exploitation is more important. For example in [43], the domain of prompt is 10k and the experiments in [43] shows that NeuralUCB is a SOTA selection strategy in this case. A comparison of MAB-FB methods and regret minimization methods in a larger domain of prompt optimization can be more helpful.

**Questions:**

What's the hyper-parameter used to run the regret minimization algorithm? e.g., UCB and NeuralUCB? Since these algorithm are highly sensitive to the selection of hyper-parameters.

**Limitations:**

See my weakness

---

> ### Author Rebuttal · Authors · 2024-08-07
>
> Thank you for reviewing this work and providing helpful comments. It is our pleasure to hear that you found the presentation is clear, the connection with BAI-FB is clean, and the experiments are extensive. To further address the raised questions and concerns, the following point-by-point response is provided.
>
> ---
>
> >**Weakness 1.** The novelty of this work is limited, since there are already a lot of prompt optimization work using bandits as the authors have discussed in the paper.
>
> **Response 1.** We would like to provide the following discussions to highlight the novelties of this work.
>
> - First and foremost, while some previous papers have touched upon leveraging bandit techniques in prompt optimization, as we have discussed in Lines 80-86, they mostly focus on using algorithms designed for **regret minimization**. As the reviewer recognized, this work makes a clean and well-justified connection between **fixed-budget best arm identification (BAI-FB)** and prompt optimization, highlighting this is a more suitable way to incorporate bandit techniques. We believe it is of great importance to clearly deliver this message.
>
> - Also, the contribution of this work goes beyond identifying the connection between BAI-FB and prompt optimization. Two enhancements, i.e., TRILE-CLST and TRIPLE-GSE, are proposed in Section 4 to leverage prompt embeddings together with BAI-FB to handle large candidate pools. Furthermore, TRIPLE-SAR and TRIPLE-CSAR are designed in Section 6 to further incorporate techniques from combinatorial bandits to efficiently perform example selection for few-shot prompts. None of these designs have been proposed in the literature of prompt optimization, and they have demonstrated superior empirical performance in the extensive experiments in Sections 5 and 6.
>
> We believe both the introduction of BAI-FB and the proposed designs contribute novel ideas to the study of prompt optimization.
>
> ---
> >**Weakness 2.** A larger domain of prompts need to be considered where exploration vs exploitation is more important. For example in [43], the domain of prompt is 10k and the experiments in [43] shows that NeuralUCB is a SOTA selection strategy in this case. A comparison of MAB-FB methods and regret minimization methods in a larger domain of prompt optimization can be more helpful.
>
> **Response 2.** During the rebuttal period, we have scaled the number of candidate prompts to $1000$. The performances on two tasks are represented as Figure 2 in the uploaded PDF. It can be observed that the proposed TRIPLE framework still achieves improved performances with this larger number of candidate prompts, further corroborating its superiority. We will add these and more results to the revised paper.
>
> Also, Figure 3 presents the performance distributions of prompt pools with sizes ranging from $100$ to $1000$. It can be observed that the distributions do not vary much with the size of the prompt pool. Thus, we believe the current experiments with $150$ and $1000$ prompts are sufficiently representative to faithfully demonstrate the compelling performance of TRIPLE.
>
> ---
> >**Question 1.** What's the hyper-parameter used to run the regret minimization algorithm? e.g., UCB and NeuralUCB? Since these algorithm are highly sensitive to the selection of hyper-parameters.
>
> **Response 3.** We used the vanilla form of UCB from [R1] where it was first proposed. For NeuralUCB, the same parameterization as the Github repo of [R2] is used. We believe this can result in a reasonable performance comparison.
>
>
> [R1] Auer, P., Cesa-Bianchi, N., and Fischer, P. (2002). Finite-time analysis of the multiarmed bandit problem.
>
> [R2] Lin, X., Wu, Z., Dai, et al. (2023). Use your instinct: Instruction optimization using neural bandits coupled with transformers.
>
> ---

---

> > ### Comment · Reviewer_cpR6 · 2024-08-13
> > **Thanks for your response**
> >
> > Thanks for your response, some of my concerns are addressed, i will keep my score.

---

> > > ### Author Response · Authors · 2024-08-13
> > >
> > > It is our pleasure to hear that the responses are helpful in addressing your concerns. We will carefully incorporate your comments in the revised paper. Thank you for recognizing the contributions of this work!

---

### Official Review · Reviewer_7VZU · 2024-07-13

**Soundness:** 2
**Presentation:** 1
**Contribution:** 2
**Rating:** 5
**Confidence:** 4

**Summary:**

This work proposes an algorithm that adopts the fixed-budget best arm (BAI-FB) identification to search for the best prompt. The authors have considered two variants of BAI-FB including sequential halving (SH) and continuously reject (CR). They then utilize prompt embeddings to enhance the BAI-FB methods via clustering and function approximation.

**Strengths:**

- The idea of using BAI-FB other than naive online MAB is interesting in prompt optimization.

- The empirical results suggest TRIPLE is an effective method to identify good prompts.

**Weaknesses:**

- The presentation could be substantially improved. For example, Section 3 seems redundant as most of its content is not critical and can be covered by Section 2. Besides, the empirical results show TRIPLE-CLST/GSE are generally better but their justifications and descriptions (including the clustering property which is critical for TRIPLE-CLST and the embedding in function approximation) are not enough in the main text. I strongly suggest authors to reduce Section 3 and extend Section 4 by adding explanations and moving some materials from the appendix into the main text.

- The evaluation setting is not described in detail. For example, are the evaluation budgets per prompt the same for all considered baselines? If so, the exact amount of evaluation budgets per prompt only means how accurate this estimation is.

- The considered prompt candidates for search are very limited only containing 150 at the maximum. However, in the literature (e.g., Lin et al. [43]), they have considered over 10k candidates. Generally, it is reasonable to consider a large prompt space for the search method to really find the optimal prompt. Otherwise, the result could be biased.

- Although the paper emphasizes efficiency, it is very hard to see if the method is efficient in the empirical section, where only the performances under the specific budget setting are considered.

- Recent baselines with/without embeddings like ZOPO and, OPRO should also be considered.

- The tasks evaluated in this paper are mostly instruction induction tasks. It would be great to show that the method can work on GLUE tasks and reasoning tasks (e.g., GSM8K, MATH).

**Questions:**

- Some literature adopts the deterministic function $f(\cdot)$ for prompt optimization and this is practical as some LLMs use greedy sampling. Will your method still work under the deterministic function $f(\cdot)$?

See other weaknesses above.

**Limitations:**

Yes.

---

> ### Author Rebuttal · Authors · 2024-08-07
>
> Thank you for reviewing this work! The following point-by-point response is provided, where the reviewer's comments are compressed due to the length limit.
>
> ---
> >**W1.** The presentation could be substantially improved...
>
> **R1.** Thank you for this helpful suggestion! In the revised paper, we will shrink Section 3, making a reduced but clear claim on the systematical connection between BAI-FB and prompt optimization. More descriptions will be added to Section 4 to highlight the empirically remarkable TRIPLE-CLST/GSE, especially the clustering property and the utilization of function approximation.
>
> ---
> >**W2.** The evaluation setting ... are the evaluation budgets per prompt the same for all considered baselines? ... evaluation budgets per prompt ...
>
> **R2.** We would like to clarify the notation of "evaluation budgets per prompt" here and in the revised paper to avoid misunderstanding.
>
> - First, as mentioned in Line 109, this work imposes an overall budget on the total interactions with LLM during training, referred to as the "(overall) budget" and denoted as $N$. The experiments always control **the overall budget as the same across methods**, ensuring fair comparisons.
>
> - The notion "evaluation budgets per prompt" is an expression denoting the value of the overall budget divided by the number of candidate prompts, i.e., $N/P$ with $P$ being the number of candidates. It just means that **on average**, each prompt is evaluated $N/P$ times, while the algorithm still can flexibly allocate the overall budget $N$. For example, in Lines 322-323, the overall budget is $N = 150$ and $P=30$ prompts are considered (thus $N/P = 5$ evaluations per prompt on average), which means the algorithm can flexibly allocate the overall $150$ evaluations on the $30$ prompts. Line 333 indicates $30$ evaluations per prompt on average, which with $P = 30$ prompts, means the overall budget is $30 \times 30 = 900$.
>
> ---
> >**W3.** The considered prompt candidates ... only containing 150 at the maximum...
>
> **R3.** During the rebuttal, we have scaled up the number of candidate prompts to $1000$, with results on two tasks reported in Figure 2 in the uploaded PDF. It can be observed that TRIPLE still exhibits better performances than others. We will add complete results to the revised paper.
>
> Also, Figure 3 demonstrates that the performance distributions of prompt pools with sizes ranging from $100$ to $1000$ are similar to each other. Thus, we believe the obtained results can faithfully represent the comparison between TRIPLE and baselines.
>
> ---
> >**W4.** ... hard to see if the method is efficient in the empirical section, where only the performances under the specific budget setting are considered.
>
> **R4.** This work refers to one method as more "efficient" than another if, under a specific (and likely stringent) budget, the former can find a better-performing prompt. We use this definition of efficiency because many previous works do not explicitly limit the LLM interactions during training, which may lead to large costs. In the empirical section, we compare the performance of the identified prompts among different methods **under the same budget** to determine which one is more efficient. We will clarify this in the revised paper.
>
> ---
> >**W5.** Recent baselines ... like ZOPO and, OPRO ...
>
> **R5.** Thank you for providing these two baselines!
>
> - [ZOPO] As the [ZOPO paper](https://arxiv.org/abs/2403.02993) is posted on arXiv on May 5th, 2024, which is very close to the NeurIPS submission deadline (full paper on May 22), we did not include it as a baseline in the initial submission. According to [NeurIPS 2024 Call For Papers](https://neurips.cc/Conferences/2024/CallForPapers#:~:text=Contemporaneous)  *"For the purpose of the reviewing process, papers that appeared online within two months of a submission will generally be considered 'contemporaneous' in the sense that the submission will not be rejected on the basis of the comparison to contemporaneous work."*, we believe our work can be considered contemporaneous with ZOPO.
>
>   Following the reviewer's helpful suggestion and [NeurIPS 2024 Call For Papers](https://neurips.cc/Conferences/2024/CallForPapers#:~:text=Contemporaneous) *"Authors are still expected to cite and discuss contemporaneous work and perform empirical comparisons to the degree feasible."*, during this rebuttal period, we have been trying our best to implement ZOPO. However, as ZOPO does not have released codes (to the best of our search) and we have encountered several clarity questions, we have not been able to re-implement it during the limited rebuttal phase. Currently, we are getting in touch with the authors to clarify our encountered issues. We will properly cite ZOPO in the revised paper and continue to do our best to add the comparisons.
>
> - [ORPO] The [ORPO paper](https://arxiv.org/abs/2309.03409), cited as [74] in our work, proposes to prompt LLMs as an optimizer to perform optimization tasks. This philosophy is very similar to that of APO [54], adopted as one pipeline in Section 5.2. Due to such a similarity, we mainly perform experiments with APO, with Table 3 demonstrating the superiority of TRIPLE.
>
> ---
> >**W6.** ... It would be great to show ... on GLUE tasks and reasoning tasks (e.g., GSM8K, MATH).
>
> **R6.** As suggested, during the rebuttal period, we have tested the performance of TRIPLE on CoLA (one of GLUE tasks) and GSM8K. Results are provided as Table 1 in the attached PDF, where TRIPLE still performs better than the baselines. We will include these and more results in the revised paper.
>
> ---
> >**Q1.** ... Will your method still work under the deterministic function $f(\cdot)$?
>
> **R7.** Yes, our solutions still work under the deterministic function, as it is one special case of the general stochastic function considered in this paper. In this case, as captured in the definition of $\mu$ in Line 97, the randomness would only come from the input $X$ but not $f(\cdot)$.
>
> ---

---

> > ### Author Response · Authors · 2024-08-13
> > **Looking forward to Discussions**
> >
> > Dear Reviewer 7VZU,
> >
> > We would like to first thank you again for the valuable comments and suggestions on our work. As the discussion period is concluding, we would be grateful if you could share any further concerns or feedback you might have. If our responses have sufficiently addressed your concerns, we hope that you could consider raising the score of your evaluation. Thank you for your consideration.
> >
> > Best,
> >
> > Authors of Submission 13481

---

> > > ### Comment · Reviewer_7VZU · 2024-08-14
> > >
> > > Thanks for the additional results and some clarifications.
> > >
> > > >More descriptions will be added to Section 4 to highlight the empirically remarkable TRIPLE-CLST/GSE
> > >
> > > As the justification and description of the method are lacking here, I am not convinced by the novelty of this work. I hope the authors can improve this part in their revision.
> > >
> > > > This philosophy is very similar to that of APO [54], adopted as one pipeline in Section 5.2. Due to such a similarity, we mainly perform experiments with APO, with Table 3 demonstrating the superiority of TRIPLE.
> > >
> > > OPRO should be a much stronger baseline than APO. I think the authors should consider the strong and representative works in this area.
> > >
> > > ---
> > > Some of my concerns are addressed, although the comparison and clarity remain an issue. I will increase my score accordingly based on the additional results provided.

---

> > > > ### Author Response · Authors · 2024-08-14
> > > >
> > > > We are glad to hear that the responses are helpful in addressing your concerns. Thank you for providing further valuable suggestions on the clarity and comparisons! We will carefully incorporate them into the revised paper, in particular additional illustrations on TRIPLE-CLST/GSE and empirical comparisons with OPRO.

---

### Official Review · Reviewer_55GB · 2024-07-30

**Soundness:** 3
**Presentation:** 4
**Contribution:** 3
**Rating:** 7
**Confidence:** 4

**Summary:**

The authors study prompt optimization, with a focus on finding the best prompt from a pool of proposed prompts under highly limited budgets. They establish a connection to fixed-budget best arm identification in multi-armed bandits (MAB) and explore several algorithms from that problem, applied to prompt selection for evaluation. They then make additional improvements by observing that prompts are not independent and that information, through embedding the prompts, can be shared to make the exploration more effective. They test methods based on clustering and DNN reward functions, based off of off-the-shelf embeddings of the prompts.

**Strengths:**

1. The paper is well-argued and well-presented. The authors argue that existing work on exploring prompt candidates under limited budgets apply a regret minimization framework, which is less well-suited than best arm identification, and then go on to explore highly-effective simple extensions that share information across prompts.

2. The results are compelling: the methods are compared against a convincing set of baselines, from UCB to Bayesian optimization (BO) with expected improvement. The outcomes consistently show the value of the author's selection of framework (BAI) and their information sharing.

**Weaknesses:**

1. The authors evaluate on tasks that to my understanding are extremely limited in scope. Most are narrow reasoning puzzles and may not reflect the type or complexity of typical prompts people use in the increasingly elaborate open-ended LM systems out there. How would such open-endedness interact with the findings?

2. The results are somewhat hard to parse from the tables and especially so from the figures, though it is clear that they are overall quite positive. The budgets considered for evaluation are perhaps very strict, which is worth calling out, e.g. <1 evaluation per prompt, 5 evaluations per prompt. The method does show gains over baselines up to 30 evalutions per prompt, so it is very compelling nonetheless.

3. The claims at the start of the paper that existing work just rarely considers budget of optimization is a bit overblown and overlL unnecessary for the claims of the paper, which are compelling on their own. The authors compare with several baselines that *do* consider budget, just not as effectively or thoughtfully as the authors do.

**Questions:**

Do TRIPLE-SAR and CSAR take ordering into account?

---

> ### Author Rebuttal · Authors · 2024-08-07
>
> Thank you for reviewing this paper and providing the helpful suggestions. We are glad to hear your recognition that this paper is well-presented and well-argued with compelling results. In the meantime, we would like to provide the following point-to-point responses, which hopefully can address your raised questions.
>
> ---
>
>
> >**Weakness 1.** The authors evaluate on tasks that to my understanding are extremely limited in scope. Most are narrow reasoning puzzles and may not reflect the type or complexity of typical prompts people use in the increasingly elaborate open-ended LM systems out there. How would such open-endedness interact with the findings?
>
> **Response 1.** We would like to answer your question in two aspects:
>
> - First, the studies on prompt optimization are focused on finding good prompts for certain downstream tasks, which contain a broad scope of applications (e.g., translate languages, generate TL;DR, summarize news, etc.). This work strictly follows this established research line, thus sharing this broad scope in terms of applications.
>
>     - Two standard datasets, Instruction-Induction and BigBench, are adopted to provide representative tasks, e.g., recommend movies based on browse histories (i.e., task "movie_recommendation" in BigBench), and translate English to German (i.e., task "translation_en-de" in Instruction-Induction).
>
>     - During the rebuttal, we have further performed experiments (see the Table 1 in the uploaded PDF) on two additional datasets, GLUE (in particular, task "Cola" on distinguishing linguistic acceptability) and GSM8K (on mathematical reasoning), to further demonstrate the broad applicability of this work. These tasks also align with other research papers on prompt optimization. We believe that our findings are convincing with these representative datasets.
>
> - Second, the line of research on prompt optimization (including our paper) can still contribute to the more open-ended interactions with LLMs mentioned by the reviewer. In particular, the identified prompts on various downstream tasks can be used to summarize general prompting strategies (e.g., use positive tongues or imperative sentences), which can be further released to guide users.
>
>   - Moreover, we note that during this summarization process, efficiency is still a major concern as good prompts on a large pool of different tasks should be identified to find the general strategies. Thus, this work contributes to providing a well-tested framework to guarantee both performance and efficiency. We will add discussions on this further extension into the revised paper.
>
> Overall, we believe this work has a sufficiently broad scope to benefit both task-specific and open-ended interactions with LLMs.
>
> ---
> >**Weakness 2.** The results are somewhat hard to parse from the tables and especially so from the figures, though it is clear that they are overall quite positive. The budgets considered for evaluation are perhaps very strict, which is worth calling out, e.g. $< 1$ evaluation per prompt, 5 evaluations per prompt. The method does show gains over baselines up to 30 evaluations per prompt, so it is very compelling nonetheless.
>
> **Response 2.** Thank you for this great suggestion on highlighting the results. We have further refined the figures and tables to facilitate understanding, emphasizing the superiority of the proposed methods. A part of the revised Figure 2(a) is contained in the uploaded PDF as Figure 1 to serve as one demonstration.
>
> - One horizontal line at 1 is added to label the baseline of Uniform, over which the performance of other methods are normalized.
> - A star is positioned on the best-performing method of each task.
> - The size of the candidate prompt pool and the stringent budget are also highlighted in the top left corner.
>
> Similar enhancements will also be applied towards other figures and tables. We believe they will better illustrate the superiority of the proposed TRIPLE framework. It would be our pleasure to further take any suggestions that you may have.
>
> ---
>
> >**Weakness 3.** The claims at the start of the paper that existing work just rarely considers budget of optimization is a bit overblown and overly unnecessary for the claims of the paper, which are compelling on their own. The authors compare with several baselines that do consider budget, just not as effectively or thoughtfully as the authors do.
>
>
> **Response 3.** Thank you for this constructive suggestion on the presentation of this work. As suggested, in the revised paper, we will put more emphasis on the obtained compelling results, especially under the strict budgets noted by the reviewer. At the meantime, a reduced while clear claim will be made that previous works did not particularly optimize performances under specific budgets, while this work introduced a systematical and empirically superior approach.
>
> ---
>
> >**Question 1.** Do TRIPLE-SAR and CSAR take ordering into account?
>
> **Response 4.** While already achieving remarkable performance, the current version of TRIPLE-SAR and CSAR do not optimize over different orderings. The main reason is that there currently lacks a comprehensive understanding on the impact of example ordering on the final performance. Directly examining all the permutations, on the other hand, would lead to an unaffordable, exponentially-large action space.
>
> If a better understanding of the impact of ordering can be established in the future (which essentially is a property of function $\mu$ defined in Line 363), it is conceivable that the TRIPLE framework can incorporate ordering by leveraging corresponding investigations on fixed-budget best arm identification, following the spirits of the pioneering TRIPLE-SAR and CSAR. We will provide further discussions on this point in the revised paper to encourage further investigations.
>
> ---

---

> > ### Comment · Reviewer_55GB · 2024-08-07
> >
> > Thank you for the detailed response. It helps me maintain my high score.

---

> > > ### Author Response · Authors · 2024-08-12
> > > **Thank you**
> > >
> > > We greatly appreciate your recognition of the contributions of this work! We will revise the paper accordingly to incorporate your suggestions.

---

### Author Rebuttal · Authors · 2024-08-07

Dear Reviewers,

Thank you for reviewing this work and giving the helpful comments! We have provided point-by-point responses, which hopefully can address the raised questions and concerns. It will be our pleasure to have further discussions and incorporate any suggestions you may have.

Together with this response, a PDF containing multiple new experimental results are uploaded, detailed in the following. We will add the complete versions of these results to the revised paper.

- [Table 1] The performance comparisons between the proposed TRIPLE framework and baselines on two new datasets, GLUE and GSM8K. The superiority of TRIPLE is further evidenced.

- [Figure 1] A sample of the improved presentation of the original Figure 2(a). We will make similar enhancements on all the figures. Let us know if you have further suggestions to improve the presentation of results.
  - One horizontal line is added to label the performance of the Uniform baseline, over which the other performances are normalized.
  - A star is added to remark the best method in each task.
  - The number of prompts and the budget in this experiment are labeled in the top left corner to facilitate references.

- [Figures 2 and 3] Further investigations on the impact of the size of the prompt pools and a new baseline BO-PI. Figure 2 demonstrates that with a large prompt pool (i.e., $1000$ prompts), TRIPLE still improves over other baselines (including the new BO-PI), illustrating its broad applicability. Figure 3 further shows that the size of the prompt pool does not have major impact on the performance distributions of its contained prompts, evidencing that the obtained results are sufficiently representative.

Thank you again and looking forward to further discussions!

Best regards,
Authors of Submission 13481

---

### Decision · Program_Chairs · 2024-09-25

**Decision:**

Accept (poster)

**Comment:**

This paper introduces TRIPLE, a framework for efficient prompt selection in large language models that optimizes performance within budget constraints by leveraging techniques from multi-armed bandit theory, demonstrating significant improvements over existing methods.

All reviewers appreciate that connecting prompt optimization to best arm identification under fixed budget is an interesting and worthwhile exploration, and is well-justified theoretically. The reviewers also appreciate the promising empirical results, supporting the authors' claim that better search algorithms are needed (and somewhat overlooked in the literature so far). On the flip side, there are questions raised on the comprehensiveness of the tasks, the performance of the algorithm with an expanded candidate prompt pool size, comparison against additional baselines, and some clarity issues that the authors were largely able to address satisfactorily during the rebuttal.

After rebuttal, all reviewers unanimously recommended the acceptance of the paper, and the AC also concurs with this recommendation after reading through reviews and subsequent author-reviewer discussions. Having said that, the AC agrees that there are several valuable suggestions from the reviewers that the authors should consider when preparing the camera-ready version, such as the discussions on more complicated and realistic tasks and toning down some rather hyperbolic claims (both made by Reviewer 55GB).